# Full-Atom Protein Pocket Design via Iterative Refinement

**Zaixi Zhang**[1,2,4], **Zepu Lu**[1,2], **Zhongkai Hao**[3], **Marinka Zitnik**[4], **Qi Liu**[1,2]*

[1] Anhui Province Key Lab of Big Data Analysis and Application,
School of Computer Science and Technology, University of Science and Technology of China
[2] State Key Laboratory of Cognitive Intelligence, Hefei, Anhui, China
[3] Dept. of Comp. Sci. and Tech., Institute for AI, THBI Lab, BNRist Center,
Tsinghua-Bosch Joint ML Center, Tsinghua, [4] Harvard University
{zaixi, zplu}@mail.ustc.edu.cn, hzj21@mails.tsinghua.edu.cn,
marinka@hms.harvard.edu, qiliuql@ustc.edu.cn

## Abstract

The design and optimization of functional proteins that bind specific ligand molecules is paramount in therapeutics and bio-engineering. A critical yet formidable task in this endeavor is the design of the protein pocket, which is the cavity region of the protein where the ligand binds. Current methods are plagued by inefficient generation, inadequate context modeling of the ligand molecule, and the inability to generate side-chain atoms. Here, we present the Full-Atom Iterative Refinement (FAIR) method, designed to address these challenges by facilitating the co-design of protein pocket sequences, specifically residue types, and their corresponding 3D structures. FAIR operates in two steps, proceeding in a coarse-to-fine manner (transitioning from protein backbone to atoms, including side chains) for a full-atom generation. In each iteration, all residue types and structures are simultaneously updated, a process termed full-shot refinement. In the initial stage, the residue types and backbone coordinates are refined using a hierarchical context encoder, complemented by two structure refinement modules that capture both inter-residue and pocket-ligand interactions. The subsequent stage delves deeper, modeling the side-chain atoms of the pockets and updating residue types to ensure sequence-structure congruence. Concurrently, the structure of the binding ligand is refined across iterations to accommodate its inherent flexibility. Comprehensive experiments show that FAIR surpasses existing methods in designing superior pocket sequences and structures, producing average improvement exceeding 10% in AAR and RMSD metrics.

## 1 Introduction

Proteins are macromolecules that perform fundamental functions in living organisms. An essential and challenging step in functional protein design, such as enzymes [52, 27] and biosensors [5, 18], is to design protein pockets that bind specific ligand molecules. Protein pockets usually refer to residues binding with the ligands in the protein-ligand complexes. Nevertheless, such a problem is very challenging due to the tremendous search space of both sequence and structure and the small solution space that satisfies the constraints of binding affinity, geometric complementarity, and biochemical stability. Traditional approaches for pocket design use hand-crafted energy functions [42, 11] and template-matching methods [67, 8, 48], which can be less accurate and lack efficiency.

---

*Qi Liu is the corresponding author.

Deep learning approaches, especially deep generative models, have made significant progress in protein design [46, 17, 9, 20] using experimental and predicted 3D structure information to design amino acid sequences [26, 30, 7, 14]. Despite their strong performance, these approaches are not directly applicable to pocket design because the protein pocket structures to be designed are apriori unknown [8]. To this end, generative models with the capability to co-design both the sequence and structure of protein pockets are needed. In another area of protein design, pioneering research developed techniques to co-design sequence and structure of complementarity determining regions (CDRs) for antibodies (Y-shaped proteins that bind with specific pathogens) [29, 28, 39, 33, 56]. Nevertheless, these approaches are tailored specifically for antibodies and are unsuitable for the pocket design of general proteins. For example, prevailing methods generate only protein backbone atoms while neglecting side-chain atoms, which play essential roles in protein pocket-ligand interactions [43]. Further, binding targets (antigens) considered by these methods are fixed, whereas binding ligand molecules are more flexible and require additional modeling.

Here, we introduce a full-atom iterative refinement approach (FAIR) for protein pocket sequence-structure co-design, addressing the limitations of prior research. FAIR consists of two primary steps: initially modeling the pocket backbone atoms to generate coarse-grained structures, followed by incorporating side-chain atoms for a complete full-atom pocket design. (1) In the first step, since the pocket residue types and the number of side-chain atoms are largely undetermined, FAIR progressively predicts and updates backbone atom coordinates and residue types via soft assignment over multiple rounds. Sequence and structure information are jointly updated in each round for efficiency, which we refer to as full-shot refinement. A hierarchical encoder with residue- and atom-level encoders captures the multi-scale structure of protein-ligand complexes. To update the pocket structure, FAIR leverages two structure refinement modules to predict the interactions within pocket residues and between the pocket and the binding ligand in each round. (2) In the second step, FAIR initializes residue types based on hard assignment and side-chain atoms based on the results of the first step. The hierarchical encoder and structure refinement modules are similar to the first step for the full-atom structure update. To model the influence of structure on residue types, FAIR randomly masks a part of the pocket residues and updates the masked residue types based on the neighboring residue types and structures in each round. After several rounds of refinement, the residue type and structure update gradually converge, and the residue sequence-structure becomes consistent. The ligand molecular structure is also updated along with the above refinement processes to account for the flexibility of the ligand.

Overall, FAIR is an E(3)-equivariant generative model that generates. both pocket residue types and the pocket-ligand complex 3D structure. Our study presents the following main contributions:

- **New task:** We preliminarily formulate protein pocket design as a co-generation of both the sequence and full-atom structure of a protein pocket (i.e., co-design), conditioned on a binding ligand molecule and the protein backbone (see Sec. 3.1).

- **Novel method:** FAIR is an end-to-end generative framework that generates both the pocket sequence and structure through iterative refinement (see Sec. 3). It addresses the limitations of previous approaches and takes into account side-chain effects, ligand flexibility, and sequence-structure consistency to enhance prediction efficiency.

- **Strong performance:** FAIR outperforms existing methods on various protein pocket design quality metrics, producing average improvements of 15.5% (AAR) and 13.5% (RMSD). FAIR's pocket-generation process is over ten times faster than traditional methods (see Sec. 4).

## 2 Related Work

**Protein Design.** Computational protein design has brought the capability of inventing proteins with desired structures and functions [22, 35, 60]. Existing studies focus on designing amino acid sequences that fold into target protein structures, i.e., inverse protein folding. Traditional approaches mainly rely on hand-crafted energy functions to iteratively search for low-energy protein sequences with heuristic sampling algorithms, which are inefficient, less accurate, and less generalizable [1, 6]. Recent deep generative approaches have achieved remarkable success in protein design [15, 26, 30, 7, 14, 51, 20, 3]. Some protein design methods also benefit from the latest advances in protein structure prediction techniques, e.g., AlphaFold2 [31]. Nevertheless, they usually assume the target protein structure is given, which is unsuitable for our pocket design case. Another

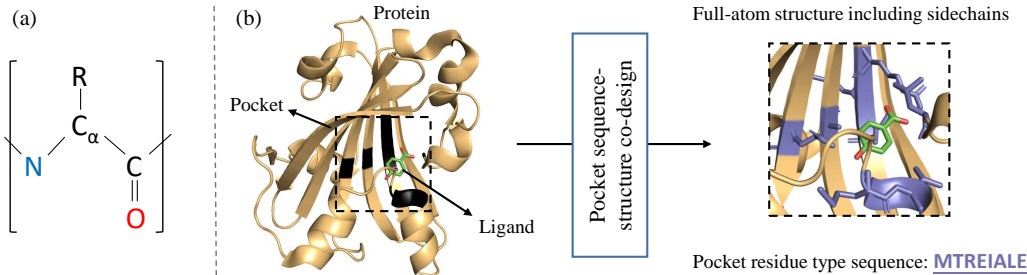

Figure 1: (a) The residue structure, where the backbone atoms are $C_\alpha, N, C, O$. $R$ represents a side chain that determines the residue types. (b) The protein pocket design problem. Pocket (colored in black) consists of a subsequence of residues closest to the binding ligand molecule (formal definition in Sec. 3.1). FAIR aims to co-design the full-atom structures and residue types of the pocket.

line of recent works proposes to co-design the sequence and 3D structure of antibodies as graphs [29, 28, 39, 33, 56]. However, they are designed explicitly for antibody CDR topologies and are primarily unable to consider side-chain atoms and the flexibility of binding targets. For more detailed discussions on protein design, interested readers may refer to the comprehensive surveys [22, 35].

**Pocket Design.** Protein pockets are the cavity region of protein where the ligand binds to the protein. In projects to design functional proteins such as biosensors [52, 27] and enzymes [5, 18], pocket design is the central task [8, 48]. Compared with the general protein design/antibody design problems, the pocket design brings unique challenges: (1) protein pocket has a smaller spatial scale than whole proteins/antibodies. Therefore, the structure and interactions of side-chain atoms play essential roles in pocket design and require careful modeling [36, 12]; (2) the influence and structural flexibility of target ligand molecules need to be considered to achieve high binding affinity [42]. Traditional methods for pocket design are physics-based methods [42, 11] or template-matching methods [67, 8, 48], which often suffer from the large computational complexity and limited accuracy. So far, the pocket design problem has rarely been tackled with deep learning.

**Structure-Based Drug Design.** Structure-based drug design is another line of research inquiry that explores 3D ligand molecule design inside protein pockets [49, 40, 37, 47, 70, 69, 19]. These studies can be regarded as the *dual* problem of pocket design studied in our paper. However, the methods primarily focus on generating 3D molecular graphs based on a fixed protein pocket structure and may not be easily adapted to the pocket sequence-structure co-design problem examined in our study. We refer interested readers to [71] for a comprehensive survey on structure-based drug design.

## 3 Full-Atom Protein Pocket Design

Next, we introduce FAIR, our method for pocket sequence-structure co-design. The problem is first formulated in Sec. 3.1. We then describe the hierarchical encoder that captures residue- and atom-level information in Sec. 3.2. Furthermore, we show the iterative refinement algorithms with two main steps in Sec. 3.3. Finally, we show the procedures for model training and sampling in Sec. 3.4 and demonstrate the E(3)-equivariance property of FAIR in Sec. 3.5.

### 3.1 Notation and Problem Statement

We aim to co-design residue types (sequences) and 3D structures of the protein pocket that can fit and bind with the target ligand molecule. As shown in Figure 1, a protein-ligand binding complex typically consists of a protein and a ligand molecule. The ligand molecule can be represented as a 3D point cloud $\mathcal{M} = \{(\boldsymbol{x}_k, \boldsymbol{v}_k)\}_{k=1}^{N_l}$ where $\boldsymbol{x}_k$ denotes the 3D coordinates and $\boldsymbol{v}_k$ denotes the atom feature. The protein can be represented as a sequence of residues $\mathcal{A} = \boldsymbol{a}_1 \cdots \boldsymbol{a}_{N_s}$, where $N_s$ is the length of the sequence. The 3D structure of the protein can be described as a point cloud of protein atoms $\{\boldsymbol{a}_{i,j}\}_{1 \le i \le N_s, 1 \le j \le n_i}$ and let $\boldsymbol{x}(\boldsymbol{a}_{i,j}) \in \mathbb{R}^3$ denote the 3D coordinate of the pocket atoms. $n_i$ is the number of atoms in a residue determined by the residue types. The first four atoms in any residue correspond to its backbone atoms $(C_\alpha, N, C, O)$, and the rest are its side-chain atoms. In our work, the protein pocket is defined as a set of residues in the protein closest to the binding ligand

molecule: $\mathcal{B} = \boldsymbol{b}_1 \cdots \boldsymbol{b}_m$. The pocket $\mathcal{B}$ can thus be represented as an amino acid subsequence of a protein: $\mathcal{B} = \boldsymbol{a}_{e_1} \cdots \boldsymbol{a}_{e_m}$ where $\boldsymbol{e} = \{e_1, \cdots, e_m\}$ is the index of the pocket residues in the protein. The index $\boldsymbol{e}$ can be formally given as: $\boldsymbol{e} = \{i \mid \min\limits_{1 \leq j \leq n_i, 1 \leq k \leq N_l} \|\boldsymbol{x}(\boldsymbol{a}_{i,j}) - \boldsymbol{x}_k\|_2 \leq \delta\}$, where $\|\cdot\|_2$ is the $L_2$ norm and $\delta$ is the distance threshold. According to the distance range of pocket-ligand interactions [43], we set $\delta = 3.5$ Å in the default setting. Note that the number of residues (i.e., $m$) varies for pockets due to varying structures of protein-ligand complexes. To summarize this subsection, our preliminary problem statement is formulated as below:

**Problem Statement:** In FAIR, our objective is to learn a conditional generative model for full-atom protein pocket generation conditioned on the protein backbone and the binding ligand. Given the availability of existing protein pockets, we may leverage them for reference to construct residue graphs and calculate geometric features. Our preliminary problem statement is formulated considering practical situations such as enzyme optimization.

## 3.2 Hierarchical Encoder

Leveraging this intrinsic multi-scale protein structure [59], we adopt a hierarchical graph transformer [44, 38, 68] to encode the hierarchical context information of protein-ligand complexes for pocket sequence-structure co-design. The transformer includes both atom- and residue-level encoders, as described below.

### 3.2.1 Atom-Level Encoder

The atom-level encoder considers all atoms in the protein-ligand complex and constructs a 3D context graph connecting the $K_a$ nearest neighboring atoms in $\mathcal{A} \bigcup \mathcal{M}$. The atomic attributes are firstly mapped to node embeddings $\boldsymbol{h}_i^{(0)}$ with two MLPs, respectively, for protein and ligand molecule atoms. The edge embeddings $\boldsymbol{e}_{ij}$ are obtained by processing pairwise Euclidean distances with Gaussian kernel functions [53]. The 3D graph transformer consists of $L$ Transformer layers [62] with two modules in each layer: a multi-head self-attention (MHA) module and a position-wise feed-forward network (FFN). Specifically, in the MHA module of the $l$-th layer ($1 \leq l \leq L$), the queries are derived from the current node embeddings $\boldsymbol{h}_i^{(l)}$ while the keys and values come from the relational representations: $\boldsymbol{r}_{ij}^{(l)} = [\boldsymbol{h}_j^{(l)}, \boldsymbol{e}_{ij}^{(l)}]$ ($[\cdot, \cdot]$ denotes concatenation) from neighboring nodes:

$$\boldsymbol{q}_i^{(l)} = \mathbf{W}_Q \boldsymbol{h}_i^{(l)}, \ \boldsymbol{k}_{ij}^{(l)} = \mathbf{W}_K \boldsymbol{r}_{ij}^{(l)}, \ \boldsymbol{v}_{ij}^{(l)} = \mathbf{W}_V \boldsymbol{r}_{ij}^{(l)}, \tag{1}$$

where $\mathbf{W}_Q, \mathbf{W}_K$ and $\mathbf{W}_V$ are learnable transformation matrices. Then, in each attention head $c \in \{1, 2, \ldots, C\}$ ($C$ is the total number of heads), the scaled dot-product attention mechanism works as follows:

$$\text{head}_i^c = \sum_{j \in \mathcal{N}(i)} \text{Softmax} \left( \frac{\boldsymbol{q}_i^{(l)^\top} \cdot \boldsymbol{k}_{ij}^{(l)}}{\sqrt{d}} \right) \boldsymbol{v}_{ij}^{(l)}, \tag{2}$$

where $\mathcal{N}(i)$ denotes the neighbors of the $i$-th atom in the constructed graph and $d$ is the dimension size of embeddings for each head. Finally, the outputs from different heads are further concatenated and transformed to obtain the final output of MHA:

$$\text{MHA}_i = \left[ \text{head}_i^1, \ldots, \text{head}_i^C \right] \mathbf{W}_O, \tag{3}$$

where $\mathbf{W}_O$ is the output transformation matrix. The output of the atom-level encoder is a set of atom representations $\{\boldsymbol{h}_i\}$. Appendix A includes the details of the atom-level encoder.

### 3.2.2 Residue-Level Encoder

The residue-level encoder only keeps the $C_\alpha$ atoms of residues and a coarsened ligand node at the ligand's center of mass to supplement binding context information (the coarsened ligand node is appended at the end of the residue sequence as a special residue). Then, a $K_r$ nearest neighbor graph is constructed at the residue level. We use the original protein pocket backbone atoms for reference to construct the $K_r$ nearest neighbor graph and help calculate geometric features. The $i$-th residue $\boldsymbol{a}_i$ can be represented by a feature vector $\boldsymbol{f}_{\boldsymbol{a}_i}^{res}$ describing its geometric and chemical characteristics such as volume, polarity, charge, hydropathy, and hydrogen bond interactions according to its residue

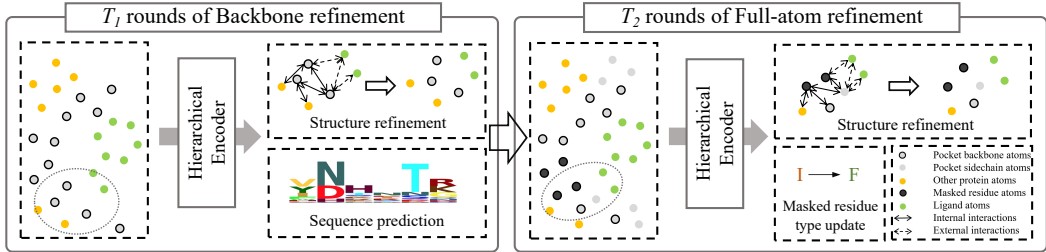

Figure 2: Overview of FAIR with two main steps ($T_1$ rounds of backbone refinement and $T_2$ rounds of full-atom refinement). FAIR co-designs pocket residue types and structures via iterative full-shot refinement. The structure refinement is illustrated with the atoms in the dotted ovals as examples.

type $\boldsymbol{a}_i$. We concatenate the residue features with the sum of atom-level embeddings $\boldsymbol{h}_k$ within each residue to initialize the residue representations; the ligand node representation is initialized by summing up all the ligand atom embeddings:

$$\hat{\boldsymbol{f}}_i^{res} = \left[ \boldsymbol{f}_{\boldsymbol{a}_i}^{res}, \sum_{j \in res_i} \boldsymbol{h}_j \right]; \ \ \hat{\boldsymbol{f}}_i^{lig} = \sum_{k \in \mathcal{M}} \boldsymbol{h}_k. \tag{4}$$

As for the edge features $\boldsymbol{e}_{ij}^{res}$, local coordinate frames are built for residues, and $\boldsymbol{e}_{ij}^{res}$ are computed according to the distance, direction, and orientation between neighboring residues [26]. Lastly, the encoder takes the node and edge features into the residue-level graph transformer to compute the residue-level representations. The residue-level graph transformer architecture is similar to that of the atom-level encoder, and details are shown in Appendix A.

In summary, the output of our hierarchical encoder is a set of residue representations $\{\boldsymbol{f}_i\}$ and atom representations $\{\boldsymbol{h}_i\}$, which are used in the following subsections for structure refinement and residue type prediction. Since our encoder is based on the atom/residue attributes and pairwise relative distances, the obtained representations are E(3)-invariant [2].

## 3.3 Iterative Refinement

For efficiency, FAIR predicts pocket residue types and structures via an iterative full-shot refinement scheme: all the pocket residue types and structures are updated together in each round. Since the pocket residue types and the number of side-chain atoms are largely unknown at the beginning rounds, FAIR is designed to have two main steps that follow a coarse-to-fine pipeline: FAIR firstly only models the backbone atoms of pockets to generate the coarse-grained structures and then fine-adjusts full-atom residues to achieve sequence-structure consistency. Figure 2 shows the overview of FAIR.

### 3.3.1 Backbone Refinement

**Initialization.** Before backbone refinement, the residue types to predict are initialized with uniform distributions over all residue type categories. Since the residue types and the number of side-chain atoms are unknown, we only initialize the 3D coordinates of backbone atoms $(C_\alpha, N, C, O)$ with linear interpolations and extrapolations based on the known structures of nearest residues in the protein. Details are shown in the Appendix A.4. Following previous research [8, 28, 33], we initialize the ligand structures with the reference structures from the dataset in the generation process.

**Residue Type Prediction.** In each round of iterative refinement, we first encode the current protein-ligand complex with the hierarchical encoder in Sec. 3.2. We then predict and update the sequences and structures. To predict the probability of each residue type, we apply an MLP and SoftMax function on the obtained pocket residue embeddings $\boldsymbol{f}_i$: $p_i = \text{Softmax}(\text{MLP}(\boldsymbol{f}_i))$, where $p_i \in \mathbb{R}^{20}$ is the predicted distribution over 20 residue categories. The corresponding residue features can then be updated with weighted averaging: $\boldsymbol{f}_{\boldsymbol{b}_i}^{res} \leftarrow \sum_j p_{i,j} \boldsymbol{f}_j^{res}$, where $p_{i,j}$ and $\boldsymbol{f}_j^{res}$ are the probability and feature vector for the $j$-th residue category respectively.

**Structure Refinement.** Motivated by the force fields in physics [50] and the successful practice in previous works [28, 33], we calculate the interactions between atoms for the coordinate update.

---

[2]E(3) is the group of Euclidean transformations: rotations, reflections, and translations.

According to previous works [32, 12, 10], the internal interactions within the protein and the external interactions between protein and ligand are different. Therefore, FAIR uses two separate modules for interaction predictions. Firstly, the internal interactions are calculated based on residue embeddings:

$$\boldsymbol{g}_{ij,kj} = g(\boldsymbol{f}_i, \boldsymbol{f}_k, \boldsymbol{e}_{\mathrm{RBF}}(\|\hat{\boldsymbol{x}}(\boldsymbol{b}_{i,1}) - \boldsymbol{x}(\boldsymbol{a}_{k,1})\|))_j \cdot \frac{\boldsymbol{x}(\boldsymbol{b}_{i,j}) - \boldsymbol{x}(\boldsymbol{a}_{k,j})}{\|\boldsymbol{x}(\boldsymbol{b}_{i,j}) - \boldsymbol{x}(\boldsymbol{a}_{k,j})\|}, \tag{5}$$

where $\boldsymbol{e}_{\mathrm{RBF}}(\|\hat{\boldsymbol{x}}(\boldsymbol{b}_{i,1}) - \boldsymbol{x}(\boldsymbol{a}_{k,1})\|)$ is the radial basis distance encoding of pairwise $C_\alpha$ distances, the function $g$ is a feed-forward neural network with four output channels for four backbone atoms, and $\frac{\boldsymbol{x}(\boldsymbol{b}_{i,j}) - \boldsymbol{x}(\boldsymbol{a}_{k,j})}{\|\boldsymbol{x}(\boldsymbol{b}_{i,j}) - \boldsymbol{x}(\boldsymbol{a}_{k,j})\|} \in \mathbb{R}^3$ is the normalized vector from the residue atom $\boldsymbol{a}_{k,j}$ to $\boldsymbol{b}_{i,j}$. For the stability of training and generation, we use original protein backbone coordinates to calculate the RBF encodings. Similarly, for the external interactions between pocket $\mathcal{B}$ and ligand $\mathcal{M}$, we have:

$$\boldsymbol{g}_{ij,s} = g'(\boldsymbol{h}_{\boldsymbol{b}_{i,j}}, \boldsymbol{h}_s, \boldsymbol{e}_{\mathrm{RBF}}(\|\boldsymbol{x}(\boldsymbol{b}_{i,j}) - \boldsymbol{x}_s\|)) \cdot \frac{\boldsymbol{x}(\boldsymbol{b}_{i,j}) - \boldsymbol{x}_s}{\|\boldsymbol{x}(\boldsymbol{b}_{i,j}) - \boldsymbol{x}_s\|}, \tag{6}$$

which is calculated based on the atom representations. $g'$ is a feed-forward neural network with one output channel. For the conciseness of expression, here we use $\boldsymbol{h}_{\boldsymbol{b}_{i,j}}$ and $\boldsymbol{h}_s$ to denote the pocket and ligand atom embeddings, respectively. Finally, the coordinates of pocket atoms $\boldsymbol{x}(\boldsymbol{b}_{i,j})$ and ligand molecule atoms $\boldsymbol{x}_s$ are updated as follows:

$$\boldsymbol{x}(\boldsymbol{b}_{i,j}) \leftarrow \boldsymbol{x}(\boldsymbol{b}_{i,j}) + \frac{1}{K_r} \sum_{k \in \mathcal{N}(i)} \boldsymbol{g}_{ij,kj} + \frac{1}{N_l} \sum_s \boldsymbol{g}_{ij,s}, \tag{7}$$

$$\boldsymbol{x}_s \leftarrow \boldsymbol{x}_s - \frac{1}{|\mathcal{B}|} \sum_{i,j} \boldsymbol{g}_{ij,s}, \ 1 \le i \le m, 1 \le j \le 4, 1 \le s \le N_l, \tag{8}$$

where $\mathcal{N}(i)$ is the neighbors of the constructed residue-level $K_r$-NN graph; $|\mathcal{B}|$ denotes the number of pocket atoms. Details of the structure refinement modules are shown in the Appendix A.

### 3.3.2 Full-Atom Refinement

**Initialization and Structure Refinement.** After rounds of backbone refinements, the predicted residue type distribution and the backbone coordinates become relatively stable (verified in experiments). We then apply the full-atom refinement procedures to determine the side-chain structures. The residue types are initialized by sampling the predicted residue type $p_i$ from backbone refinement (hard assignment). The number of side-chain atoms and the atom types can then be determined. The coordinates of the side-chain atoms are initialized with the corresponding $\alpha$-carbon coordinates. The full-atom structure refinement is similar to backbone refinement (Sec. 3.3.1) with minor adjustments to accommodate the side-chain atoms. We include the details in the Appendix A.

**Residue Type Update.** Since the structures and interactions of side-chain atoms can further influence the residue types, proper residue type update algorithms are needed to achieve sequence-structure consistency. Considering the hard assignment of residue types in the full-atom refinement, we randomly sample and mask one residue in each protein pocket and update its residue type based on the surrounding environment in each iteration. Specifically, we mask the residue types, remove the sampled residues' side-chain atoms in the input to the encoder, and use the obtained corresponding residue types for residue type prediction. If the predicted result is inconsistent with the original residue type, we will update the residue type with the newly predicted one and reinitialize the side-chain structures. As our experiments indicate, the residue types and structures become stable after several refinement iterations, and sequence-structure consistency is achieved.

### 3.4 Model Training and Sampling

The loss function for FAIR consists of two parts. We apply cross-entropy loss $l_{ce}$ for (sampled) pocket residues over all iterations for the sequence prediction. For the structure refinement, we adopt the Huber loss $l_{huber}$ for the stability of optimization following previous works [28, 56]:

$$\mathcal{L}_{seq} = \frac{1}{T} \sum_t \sum_i l_{ce}(p_i^t, \hat{p}_i); \tag{9}$$

$$\mathcal{L}_{struct} = \frac{1}{T} \sum_t \left[ \sum_i l_{huber}(\boldsymbol{x}(\boldsymbol{b}_i)^t, \hat{\boldsymbol{x}}(\boldsymbol{b}_i)) + \sum_j l_{huber}(\boldsymbol{x}_j^t, \hat{\boldsymbol{x}}_j) \right], \tag{10}$$

where $T = T_1 + T_2$ is the total rounds of refinement. $\hat{p}_i, \hat{\boldsymbol{x}}(\boldsymbol{b}_i)$, and $\hat{\boldsymbol{x}}_j$ are the ground-truth residue types, residue coordinates, and ligand coordinates. $p_i^t, \boldsymbol{x}(\boldsymbol{b}_i)^t$, and $\boldsymbol{x}_j^t$ are the predicted ones at the $t$-th round. In the training process, we aim to minimize the sum of the above two loss functions $\mathcal{L} = \mathcal{L}_{seq} + \mathcal{L}_{struct}$ similar to previous works [56, 34]. After the training of FAIR, we can co-design pocket sequences and structures. Algorithm 1 and 2 in Appendix A outline model training and sampling.

### 3.5 Equivariance

FAIR is E(3)-equivariant, which is a desirable property of generative protein models [33, 64]:

**Theorem 1.** *Denote the E(3)-transformation as $T_g$ and the generative process of FAIR as $\{(\boldsymbol{b}_i, \boldsymbol{x}(\boldsymbol{b}_i))\}_{i=1}^m = p_\theta(\mathcal{A} \setminus \mathcal{B}, \mathcal{M})$, where $\{(\boldsymbol{b}_i, \boldsymbol{x}(\boldsymbol{b}_i))\}_{i=1}^m$ indicates the designed pocket sequence and structure. We have $\{(\boldsymbol{b}_i, T_g \cdot \boldsymbol{x}(\boldsymbol{b}_i))\}_{i=1}^m = p_\theta(T_g \cdot (\mathcal{A} \setminus \mathcal{B}), T_g \cdot \mathcal{M})$.*

*Proof.* The main idea is that the E(3)-invariant encoder and E(3)-equivariant structure refinement lead to an E(3)-equivariant generative process of FAIR. We give the full proof in Appendix A. □

## 4 Experiments

We proceed by describing the experimental setups (Sec. 4.1). We then assess our method for ligand-binding pocket design (Sec. 4.2) on two datasets. We also perform further in-depth analysis of FAIR, including sampling efficiency analysis and ablations in Sec. 4.3.

### 4.1 Experimental Setting

**Datasets.** We consider two widely used datasets for experimental evaluations. **CrossDocked** dataset [16] contains 22.5 million protein-molecule pairs generated through cross-docking. We filter out data points with binding pose RMSD greater than 1 Å, leading to a refined subset with around 180k data points. For data splitting, we use mmseqs2 [58] to cluster data at 30% sequence identity, and randomly draw 100k protein-ligand structure pairs for training and 100 pairs from the remaining clusters for testing and validation, respectively. **Binding MOAD** dataset [21] contains around 41k experimentally determined protein-ligand complexes. We further filter and split the Binding MOAD dataset based on the proteins' enzyme commission number [4], resulting in 40k protein-ligand pairs for training, 100 pairs for validation, and 100 pairs for testing following previous work [54]. More data split schemes based on graph clustering [66, 65] may be explored.

Considering the distance ranges of protein-ligand interactions [43], we redesign all the residues that contain atoms within 3.5 Å of any binding ligand atoms, leading to an average of around eight residues for each pocket. We sample 100 sequences and structures for each protein-ligand complex in the test set for a comprehensive evaluation.

**Baselines and Implementation.** FAIR is compared with five state-of-the-art representative baseline methods. **PocketOptimizer** [45] is a physics-based method that optimizes energies such as packing and binding-related energies for ligand-binding protein design. **DEPACT** [8] is a template-matching method that follows a two-step strategy [67] for pocket design. It first searches the protein-ligand complexes in the database with similar ligand fragments. It then grafts the associated residues into the protein pocket with PACMatch [8]. **HSRN** [28], **Diffusion** [39], and **MEAN** [33] are deep-learning-based models that use auto-regressive refinement, diffusion model, and graph translation method respectively for protein sequence-structure co-design. They were originally proposed for antibody design, and we adapted them to our pocket design task with proper modifications (see Appendix. B for more details).

We use AMBER ff14S force field [41] for energy computation and the Dunbrack rotamer library [55] for rotamer sampling in PocketOptimizer. For the template-based method DEPACT, the template databases are constructed based on the training datasets for fair comparisons. For the deep-learning-based models, including our FAIR, we train them for 50 epochs and select the checkpoint with the lowest loss on the validation set for testing. We use the Adam optimizer with a learning rate of 0.0001 for optimization. In FAIR, the default setting sets $T_1$ and $T_2$ as 5 and 10. Descriptions of baselines and FAIR are provided in the Appendix. B.

Table 1: Evaluation of different approaches on the pocket design task.

| Model | CrossDocked | | | Binding MOAD | | |
|---|---|---|---|---|---|---|
| | AAR (↑) | RMSD (↓) | Vina (↓) | AAR (↑) | RMSD (↓) | Vina (↓) |
| PocketOptimizer | 27.89±14.9% | 1.75±0.08 | -6.905±2.39 | 28.78±11.3% | 1.68±0.12 | -7.829±2.41 |
| DEPACT | 22.58±8.48% | 1.97±0.14 | -6.670±2.13 | 26.12±8.97% | 1.76±0.15 | -7.526±2.05 |
| HSRN | 31.62±10.4% | 2.15±0.17 | -6.565±1.95 | 33.70±10.1% | 1.83±0.18 | -7.349±1.93 |
| Diffusion | 34.62±13.7% | 1.68±0.12 | -6.725±1.83 | 36.94±12.9% | 1.47±0.09 | -7.724±2.36 |
| MEAN | 35.46±8.15% | 1.76±0.09 | -6.891±1.86 | 37.16±14.7% | 1.52±0.09 | -7.651±1.97 |
| FAIR | **40.17±12.6%** | **1.42±0.07** | **-7.022±1.75** | **43.75±15.2%** | **1.35±0.10** | **-7.978±1.91** |

**Performance Metrics.** To evaluate the generated sequences and structures, we employ Amino Acid Recovery (**AAR**), defined as the overlapping ratio between the predicted 1D sequences and ground truths, and Root Mean Square Deviation (**RMSD**) regarding the 3D predicted structure of residue atoms. Due to the different residue types and numbers of side-chain atoms in the generated pockets, we calculate RMSD with respect to backbone atoms following [28, 33]. To measure the binding affinity between the protein pocket and ligand molecules, we calculate **Vina Score** with QVina [61, 2]. The unit is kcal/mol; a lower Vina score indicates better binding affinity. Before feeding to Vina, all the generated protein structures are firstly refined by AMBER force field [41] in OpenMM [13] to optimize the structures. For the baseline methods that predict backbone structures only (Diffusion and MEAN), we use Rosetta [1] to do side-chain packing following the instructions in their papers.

## 4.2 Ligand-Binding Pocket Design

**Comparison with Baselines.** Table 1 shows the results of different methods on the CrossDocked and Binding MOAD dataset. FAIR overperforms previous methods with a clear margin on AAR, RMSD, and Vina scores, which verifies the strong ability of FAIR to co-design pocket sequences and structures with high binding affinity. For example, the average improvements on AAR and RMSD are 15.5% and 13.5% respectively. Compared with the energy-based method PocketOptimizer and the template-matching-based method DEPACT, deep-learning-based methods such as FAIR have better performance due to their stronger ability to model the complex sequence-structure dependencies and pocket-ligand interactions. However, these deep-learning-based methods still have limitations that restrict their performance improvement. For example, the autoregressive generation manner in HSRN inevitably incurs error accumulation; Diffusion and MEAN fail to model the influence of side chain atoms; all these methods ignore the flexibility of ligands. In contrast, FAIR addresses the limitations with properly designed modules, further validated in Sec. 4.3. Moreover, FAIR performs relatively better on Binding MOAD than the CrossDocked dataset, which may be explained by the higher quality and diversity of Binding MOAD [21] (e.g. Vina score on CrossDocked -7.022 vs. -7.978 on Binding MOAD). For example, Binding MOAD has 40k unique protein-ligand complexes, while CrossDocked only contains 18k complexes.

**Comparison of Generated and Native Sequences.** Figure 4(a) shows the residue type distributions of the designed pockets and the native ones from two datasets. Generally, we observe that the distributions align very well. Some residue types, such as Tyrosine and Asparagine, have been used more frequently in designed than naturally occurring sequences. In contrast, residue types with obviously reduced usages in the designed sequences include Lysine, Isoleucine, and Leucine.

**Convergence of Iterative Refinement Process.** During the iterations of FAIR, we monitor the ratio of unconverged residues (residue types different from those in the final designed sequences) and AAR, shown in Figure 4(b) ($T_1 = 5$ and $T_2 = 15$). We observe that both curves converge quickly during refinement. We find that the AAR becomes stable in just several iterations (typically less than 3 iterations in our experiments) of backbone refinement and then gradually increases with the complete atom refinement, validating the effectiveness of our design. We show the curves of RMSD in Appendix C.3, which also converges rapidly. Therefore, we set $T_1 = 5$ and $T_2 = 10$ for efficiency as default setting.

**Case Studies.** In Figure. 3, we visualize two examples of designed pocket sequences and structures from the CrossDocked dataset (PDB: 2pc8) and the Binding MOAD dataset (PDB: 6p7r), respectively. The designed sequences recover most residues, and the generated structures are valid and realistic. Moreover, the designed pockets exhibit comparable or higher binding affinities (lower Vina scores) with the target ligand molecules than the reference pockets in the test set.

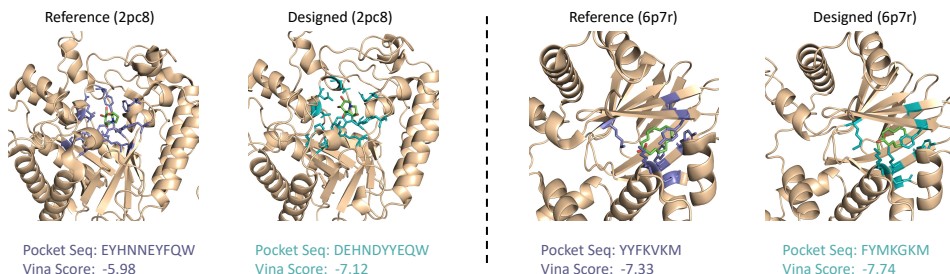

| Reference (2pc8) | Designed (2pc8) | Reference (6p7r) | Designed (6p7r) |
|---|---|---|---|
| Pocket Seq: EYHNNEYFQW | Pocket Seq: DEHNDYYEQW | Pocket Seq: YYFKVKM | Pocket Seq: FYMKGKM |
| Vina Score: -5.98 | Vina Score: -7.12 | Vina Score: -7.33 | Vina Score: -7.74 |

Figure 3: Case studies of Pocket design. We show the reference and designed structures/sequences of two protein pockets from the CrossDocked (PDB: 2pc8) and Binding MOAD datasets (PDB: 6p7r).

### 4.3 Additional Analysis of FAIR's Performance

**Sampling Efficiency Analysis.** To evaluate the efficiency of FAIR, we considered the generation time of different approaches using a single V100 GPU on the same machine. Average runtimes are shown in Figure 4(c). First, the generation time of FAIR is more than ten times faster (FAIR: 5.6s vs. DEPACT: 107.2s) than traditional PocketOptimizer and DEPACT methods. Traditional methods are time-consuming because they require explicit energy calculations and combinatorial enumerations. Second, FAIR is more efficient than the autoregressive HSRN and diffusion methods. The MEAN sampling time is comparable to FAIR since both use full-shot decoding schemes. However, FAIR has the advantage of full-atom prediction over MEAN.

**Local Geometry of Generated Pockets.** Since RMSD values in Tables 1 and 2 reflect only the global geometry of generated protein pockets, we also consider the RMSD of bond lengths and angles of predicted pockets to evaluate their local geometry [33]. We measure the average RMSD of the covalent bond lengths in the pocket backbone (C-N, C=O, and C-C). We consider three conventional dihedral angles in the backbone structure, i.e., $\phi, \psi, \omega$ [57] and calculate the average RMSD of their cosine values. The results in Table 3 of Appendix C.3 show that FAIR achieves better performance regarding the local geometry than baseline methods. The RMSD of bond lengths and angles drop 20% and 8% respectively with FAIR.

**Ablations.** In Table 2, we evaluate the effectiveness and necessity of the proposed modules in FAIR. Specifically, we removed the residue-level encoder, the masked residue type update procedure, the full-atom refinement, and the ligand molecule context, denoted as w/o the encoder/consistency/full-atom/ligand. The results in Table 2 demonstrate that the removal of any of these modules leads to a decrease in performance. We observe that the performance of FAIR w/o hierarchical encoder drops the most. This is reasonable, as the hierarchical encoder captures essential representations for sequence/structure prediction. Moreover, the masked residue type update procedure helps ensure sequence-structure consistency. Additionally, the full-atom refinement module and the binding ligand are essential for modeling side-chain and pocket-ligand interactions, which play important roles in protein-ligand binding.

Further, we replaced the iterative full-shot refinement modules in FAIR with the autoregressive decoding method used in HSRN [28] as a variant of FAIR. However, this variant's performance is also worse than FAIR's. As discussed earlier, autoregressive generation can accumulate errors and is more time-consuming than the full-shot refinement in FAIR. For example, AAR drops from 40.17% to 31.04% on CrossDocked with autoregressive generation.

Table 2: Ablation studies of FAIR.

| Model | CrossDocked | | | Binding MOAD | | |
|---|---|---|---|---|---|---|
| | AAR (↑) | RMSD (↓) | Vina (↓) | AAR (↑) | RMSD (↓) | Vina (↓) |
| w/o hier encoder | 25.30±9.86% | 2.36±0.24 | -6.694±2.28 | 29.21±10.8% | 1.82±0.14 | -7.235±2.10 |
| w/o consistency | 36.75±9.21% | 1.49±0.08 | -6.937±1.72 | 37.40±15.0% | 1.49±0.12 | -7.752±1.94 |
| w/o full-atom | 35.52±10.4% | 1.62±0.13 | -6.412±2.37 | 35.91±13.5% | 1.54±0.12 | -7.544±2.23 |
| w/o ligand | 33.16±15.3% | 1.71±0.11 | -6.353±1.91 | 34.54±14.6% | 1.50±0.09 | -7.271±1.79 |
| autoregressive | 31.04±14.2% | 1.56±0.16 | -6.874±1.90 | 33.65±12.7% | 1.46±0.11 | -7.765±1.84 |
| **FAIR** | **40.17±12.6%** | **1.42±0.07** | **-7.022±1.75** | **43.75±15.2%** | **1.35±0.10** | **-7.978±1.91** |

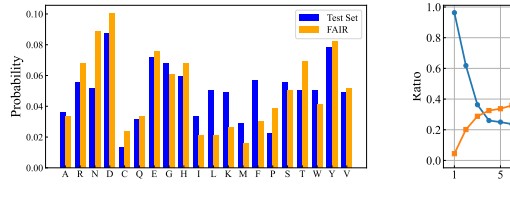 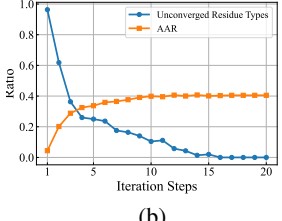 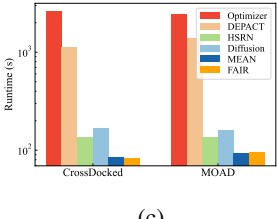

| (a) | (b) | (c) |

Figure 4: (a) The distributions of residue types in the designed pockets and the test sets. (b) The ratio of unconverged residues (residue types different from those in the final designed sequences) and AAR (Amino Acid Recovery) during the iterations of FAIR on the CrossDocked dataset. (c) Comparisons of average generation time for 100 pockets with baseline methods.

## 5  Limitations and Future Research

While we have made significant strides in addressing the limitations of prior research, several avenues remain for future research. First, we preliminarily formulate the protein pocket co-design problem in this paper. In the future, more realistic problem formulations may be constructed according to the requirements of different scenarios. Second, beyond the design of protein pockets, optimizing properties such as the binding affinity of existing pockets is crucial, especially for therapeutic applications. In forthcoming research, we aim to combine reinforcement learning and Bayesian optimization with FAIR to refine pocket design. Third, leveraging physics-based and template-matching methodologies can pave the way for the development of hybrid models that are both interpretable and generalizable. For instance, physics-based techniques can be employed to sample various ligand conformations during ligand initialization. The inherent protein-ligand interaction data within templates can guide pocket design. Lastly, integrating FAIR into protein/enzyme design workflows has the potential to be immensely beneficial for the scientific community. Conducting wet-lab experiments to assess the efficacy of protein pockets designed using FAIR would be advantageous. Insights gleaned from these experimental outcomes can be instrumental in refining our model, thereby establishing a symbiotic relationship between computation and biochemical experiments.

## 6  Conclusion

We develop a full-atom iterative refinement framework (FAIR) for protein pocket sequence and 3D structure co-design. Generally, FAIR has two refinement steps (backbone refinement and full-atom refinement) and follows a coarse-to-fine pipeline. The influence of side-chain atoms, the flexibility of binding ligands, and sequence-structure consistency are well considered and addressed. We empirically evaluate our method on cross-docked and experimentally determined datasets to show the advantage over existing physics-based, template-matching-based, and deep generative methods in efficiently generating high-quality pockets. We hope our work can inspire further explorations of pocket design with deep generative models and benefit the broad community.

## 7  Acknowledgements

This research was partially supported by grants from the National Key Research and Development Program of China (No. 2021YFF0901003) and the University Synergy Innovation Program of Anhui Province (GXXT-2021-002). We wish to express our sincere appreciation to Dr. Yaoxi Chen and Yifei Hu for their constructive discussions, which greatly enrich this research.

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

# A Details on FAIR Model

## A.1 Training and Sampling Algorithms

Here, we show the pseudo-codes of the training and sampling of FAIR in the Algorithm. 1 and 2.

---

**Algorithm 1** Training of FAIR

---

**Input:** protein sequences $\mathcal{A} = \boldsymbol{a}_1 \cdots \boldsymbol{a}_{N_s}$ and structures $\{\boldsymbol{x}(\boldsymbol{a}_i)\}_{i=1}^{N_s}$, ligand molecule $\mathcal{M}$.
Initialize the coordinates of pocket residues $\{\boldsymbol{x}(\boldsymbol{b}_i)\}$.
Add Gaussian noise to the ligand molecular structure $\{\boldsymbol{x}_j\}$.
Initialize the sequences of pocket residues with uniform distributions over 20 residue categories.
$\mathcal{L}_{pred} = 0; \mathcal{L}_{struct} = 0$.
\# Backbone refinement
**for** $t$ in $1, \cdots T_1$:
    Obtain the residue embeddings $\{\boldsymbol{f}_i\}$ and atom embeddings $\{\boldsymbol{h}_i\}$ with the hierarchical encoder.
    Predict residue types and update coordinates with structure refinement modules.
    $\mathcal{L}_{seq} += \sum_i l_{ce}(p_i^t, \hat{p}_i)$.
    $\mathcal{L}_{struct} += \sum_i l_{huber}(\boldsymbol{x}(\boldsymbol{b}_i)^t, \hat{\boldsymbol{x}}(\boldsymbol{b}_i)) + \sum_j l_{huber}(\boldsymbol{x}_j^t, \hat{\boldsymbol{x}}_j)$.
**end for**
Sample residue types from the predicted residue type distributions.
Initialize the coordinates of side-chain atoms with the coordinates of corresponding $C_\alpha$.
\# Full-atom refinement
**for** $t$ in $1, \cdots T_2$:
    Randomly mask a subset of pocket residues.
    Obtain the residue embeddings $\{\boldsymbol{f}_i\}$ and atom embeddings $\{\boldsymbol{h}_i\}$ with the hierarchical encoder.
    Predict the residue types of masked residues.
    $\mathcal{L}_{seq} += \sum_i l_{ce}(p_i^t, \hat{p}_i)$.
    Reinitialize the masked residue if its residue type conflicts with the predicted result.
    Update coordinates with structure refinement modules.
    $\mathcal{L}_{struct} += \sum_i l_{huber}(\boldsymbol{x}(\boldsymbol{b}_i)^t, \hat{\boldsymbol{x}}(\boldsymbol{b}_i)) + \sum_j l_{huber}(\boldsymbol{x}_j^t, \hat{\boldsymbol{x}}_j)$.
**end for**
Minimize $\frac{1}{T}(\mathcal{L}_{seq} + \mathcal{L}_{struct})$.

---

---

**Algorithm 2** Sampling of FAIR

---

**Input:** protein sequences $\mathcal{A} = \boldsymbol{a}_1 \cdots \boldsymbol{a}_{N_s}$ and structures $\{\boldsymbol{x}(\boldsymbol{a}_i)\}_{i=1}^{N_s}$, ligand molecule $\mathcal{M}$.
Initialize the coordinates of pocket residues $\{\boldsymbol{x}(\boldsymbol{b}_i)\}$.
Initialize the sequences of pocket residues with uniform distributions over 20 residue categories.
\# Backbone refinement
**for** $t$ in $1, \cdots T_1$:
    Obtain the residue embeddings $\{\boldsymbol{f}_i\}$ and atom embeddings $\{\boldsymbol{h}_i\}$ with the hierarchical encoder.
    Predict residue types and update coordinates with structure refinement modules.
**end for**
Sample residue types from the predicted residue type distributions.
Initialize the coordinates of side-chain atoms with the coordinates of corresponding $C_\alpha$.
\# Full-atom refinement
**for** $t$ in $1, \cdots T_2$:
    Randomly mask a subset of pocket residues.
    Obtain the residue embeddings $\{\boldsymbol{f}_i\}$ and atom embeddings $\{\boldsymbol{h}_i\}$ with the hierarchical encoder.
    Predict the residue types of masked residues.
    Reinitialize the masked residue if its residue type conflicts with the predicted result.
    Update coordinates with structure refinement modules.
**end for**

---

## A.2 Additional Information on Hierarchical Graph Transformer Encoder

Our hierarchical encoder includes the atom-level encoder and the residue-level encoder. The powerful 3D graph transformer is used as the model backbone. Here, we first provide additional information on the graph transformer. Residue-level node and edge feature representations follow prior research [26, 28].

**Graph transformer architecture.** The atom/residue-level encoder contains $L$ graph transformer layers. Let $h^{(l)}$ be the set of node representations at the $l$-th layer. In each graph transformer layer, there is a multi-head self-attention (MHA) and a feed-forward block (FFN). The layer normalization (LN) is applied before the two blocks [63, 68]. The details of MHA have been shown in Sec 3.2, and the graph transformer layer is formally characterized as:

$$h'^{(l-1)} = \text{MHA}(\text{LN}(h^{(l-1)})) + h^{(l-1)} \tag{11}$$

$$h^{(l)} = \text{FFN}(\text{LN}(h'^{(l-1)})) + h'^{(l-1)}, \quad (0 \leq l < L). \tag{12}$$

**Residue-level node features.** The residue-level encoder only keeps the $C_\alpha$ atoms to represent residues and constructs a $K_r$ nearest neighbor graph at the residue level. We use the original protein pocket backbone atoms for reference to construct the $K_r$ nearest neighbor graph and help calculate geometric features. Each residue node is represented by six features: polarity $f_p \in \{0, 1\}$, hydropathy $f_h \in [-4.5, 4.5]$, volume $f_v \in [60.1, 227.8]$, charge $f_c \in \{-1, 0, 1\}$, and whether it is a hydrogen bond donor $f_d \in \{0, 1\}$ or acceptor $f_a \in \{0, 1\}$. We expand hydropathy and volume features into radial basis with interval sizes 0.1 and 10, respectively. Overall, the dimension of the residue-level feature vector $f_i$ is 112.

**Residue-level edge features.** For the $i$-th residue, we let $x(a_{i,1})$ denote the coordinate of its $C_\alpha$ and define its local coordinate frame $O_i = [c_i, n_i, c_i \times n_i]$ as:

$$u_i = \frac{x(a_{i,1}) - x(a_{i-1,1})}{\|x(a_{i,1}) - x(a_{i-1,1})\|}, \quad c_i = \frac{u_i - u_{i+1}}{\|u_i - u_{i+1}\|}, \quad n_i = \frac{u_i \times u_{i+1}}{\|u_i \times u_{i+1}\|}. \tag{13}$$

Based on the local frame, the edge features between residues $i$ and $j$ can be computed as:

$$e_{ij}^{res} = \text{Concat}\left(E_{\text{pos}}(i-j), \, e_{\text{RBF}}(\|x(a_{i,1}) - x(a_{j,1})\|), \, O_i^\top \frac{x(a_{j,1}) - x(a_{i,1})}{\|x(a_{i,1}) - x(a_{j,1})\|}, \, q(O_i^\top O_j)\right). \tag{14}$$

The edge feature $e_{ij}^{res}$ contains four parts. The positional encoding $E_{\text{pos}}(i - j)$ encodes the relative sequence distance between two residues. The second term $e_{\text{RBF}}(\cdot)$ is a distance encoding with radial basis functions. The third term is a direction encoding corresponding to the relative direction of $x(a_{j,1})$ in the local frame of the $i$-th residue. The last term $q(O_i^\top O_j)$ is the orientation encoding of the quaternion representation $q(\cdot)$ of the spatial rotation matrix $O_i^\top O_j$ [25]. Overall, the dimension of the residue-level edge feature $e_{ij}^{res}$ is 39.

## A.3 More Details of the Full-atom Structure Refinement

The full-atom structure refinement is similar to the backbone structure refinement in Sec. 3.3.1 with minor modifications to accommodate side-chain atoms. We consider the internal interactions within the protein and the external interactions between ligand molecules and protein residues. Firstly, the internal interactions within the protein are calculated based on residue embeddings:

$$g_{ij,kj} = g(f_i, f_k, e_{\text{RBF}}(\|\hat{x}(b_{i,1}) - x(a_{k,1})\|))_j \cdot \frac{x(b_{i,j}) - \bar{x}(a_k)}{\|x(b_{i,j}) - \bar{x}(a_k)\|}, \tag{15}$$

where $e_{\text{RBF}}(\|\hat{x}(b_{i,1}) - x(a_{k,1})\|)$ is the radial basis distance encoding of pairwise $C_\alpha$ distances, the function $g$ is a feed-forward neural network with 14 channels for all side-chain atoms (the maximum number of atoms in a residue is 14). Due to the different numbers of side-chain atoms in different kinds of residues, the normalized direction vector here is $\frac{x(b_{i,j}) - \bar{x}(a_k)}{\|x(b_{i,j}) - \bar{x}(a_k)\|} \in \mathbb{R}^3$ where $\bar{x}(a_k)$ is the average coordinates of residue $a_k$. The external interactions between pocket $\mathcal{B}$ and ligand $\mathcal{M}$ is the same as the backbone refinement:

$$g_{ij,s} = g'(h_{b_{i,j}}, h_s, e_{\text{RBF}}(\|x(b_{i,j}) - x_s\|)) \cdot \frac{x(b_{i,j}) - x_s}{\|x(b_{i,j}) - x_s\|}, \tag{16}$$

which is calculated based on the atom representations. For the conciseness of expression, here we use $\boldsymbol{h}_{\boldsymbol{b}_{i,j}}$ and $\boldsymbol{h}_s$ to denote the pocket and ligand atom embeddings, respectively. Finally, the coordinates of pocket atoms $\boldsymbol{x}(\boldsymbol{b}_{i,j})$ and ligand molecule atoms $\boldsymbol{x}_s$ are updated as follows:

$$\boldsymbol{x}(\boldsymbol{b}_{i,j}) \leftarrow \boldsymbol{x}(\boldsymbol{b}_{i,j}) + \frac{1}{K_r} \sum_{k \in \mathcal{N}(i)} \boldsymbol{g}_{ij,kj} + \frac{1}{N_l} \sum_s \boldsymbol{g}_{ij,s}, \tag{17}$$

$$\boldsymbol{x}_s \leftarrow \boldsymbol{x}_s - \frac{1}{|\mathcal{B}|} \sum_{i,j} \boldsymbol{g}_{ij,s},\ 1 \le i \le m, 1 \le j \le n_i, 1 \le s \le N_l, \tag{18}$$

where $\mathcal{N}(i)$ indicates the neighbors in the constructed residue-level $K_r$-NN graph; $|\mathcal{B}|$ denotes the number of pocket atoms.

## A.4 Structure initialization based on interpolation and extrapolation

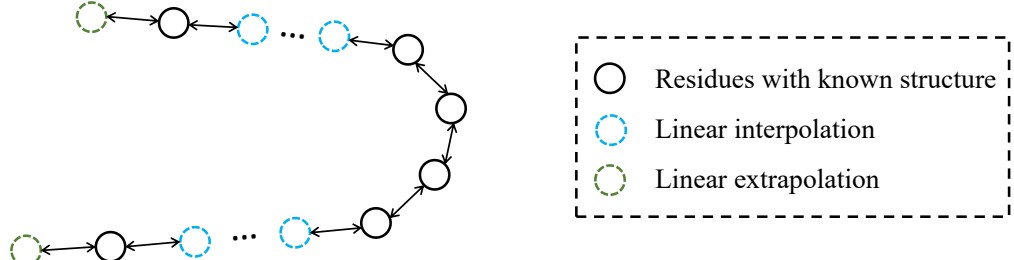

Figure 5: Structure initialization based on interpolation and extrapolation

Figure 5 illustrates our structure initialization strategy. We initialize the residue coordinates with linear interpolations and extrapolations based on the nearest residues with known structures in the protein. Denote the sequence of residues as $\mathcal{A} = \boldsymbol{a}_1 \cdots \boldsymbol{a}_{N_s}$, where $N_s$ is the length of the sequence. Let $\boldsymbol{Z}(\boldsymbol{a}_i) \in \mathbb{R}^{3 \times 4}$ denote the backbone coordinates of the $i$-th residue. If we want to initialize the coordinates of the $i$-th residue, we take the following initialization strategy: (1) We use linear interpolation if there are residues with known coordinates at both sides of the $i$-th residue. Specifically, assume $p$ and $q$ are the indexes of the nearest residues with known coordinates at each side of the $i$-th residue ($p < i < q$), we have: $\boldsymbol{Z}(\boldsymbol{a}_i) = \frac{1}{q-p}[(i-p)\boldsymbol{Z}(\boldsymbol{a}_q) + (q-i)\boldsymbol{Z}(\boldsymbol{a}_p)]$. (2) We conduct linear extrapolation if the $i$-th residue is at the ends of the chain, i.e., no residues with known structures at one side of the $i$-th residue. Specifically, let $p$ and $q$ denote the index of the nearest and the second nearest residue with known coordinates. The position of the $i-$th residue can be initialized as $\boldsymbol{Z}(\boldsymbol{a}_i) = \boldsymbol{Z}(\boldsymbol{a}_p) + \frac{i-p}{p-q}(\boldsymbol{Z}(\boldsymbol{a}_p) - \boldsymbol{Z}(\boldsymbol{a}_q))$. We initialize the side-chain atoms' coordinates with the coordinate of their corresponding $C_\alpha$. In practice, we can further apply local geometric constraints to the initialized backbone atoms in each residue.

## A.5 Equivariance.

FAIR has the desirable property of E(3)-equivariance as follow:

**Theorem 1.** *Denote the E(3)-transformation as $T_g$ and the generative process of FAIR as $\{(\boldsymbol{b}_i, \boldsymbol{x}(\boldsymbol{b}_i))\}_{i=1}^m = p_\theta(\mathcal{A} \setminus \mathcal{B}, \mathcal{M})$, where $\{(\boldsymbol{b}_i, \boldsymbol{x}(\boldsymbol{b}_i))\}_{i=1}^m$ indicates the designed pocket seqeuce and structure. We have $\{(\boldsymbol{b}_i, T_g \cdot \boldsymbol{x}(\boldsymbol{b}_i))\}_{i=1}^m = p_\theta(T_g \cdot (\mathcal{A} \setminus \mathcal{B}), T_g \cdot \mathcal{M})$.*

The E(3)-transformation on the euclidean coordinate $\boldsymbol{x} \in \mathbb{R}^3$ can be represented as: $T_g \cdot \boldsymbol{x} = O\boldsymbol{x} + t$, where $O \in \mathbb{R}^3$ is the orthogonal transformation matrix, $t \in \mathbb{R}^3$ is the translation vector. Before we give the proof, we present the following lemmas.

**Lemma 1.** *The hierarchical graph transformer encoder is E(3)-invariant.*

*Proof.* As shown in Sec. 3.2, the input to the hierarchical graph transformer encoders are scalar node features and pairwise distances, and the output is atom and residue scalar features. Therefore, the hierarchical encoder is E(3)-invariant by construction. □

**Lemma 2.** *The backbone and full-atom structure refinement are E(3)-equivariant.*

*Proof.* Firstly, with the E(3)-invariant encoder, the obtained residue embeddings $\{\boldsymbol{f}_i\}$, atom embeddings $\{\boldsymbol{h}_i\}$, and the output of functions $g$ and $g'$ are E(3)-invariant. Furthermore, $\forall T_g \in \mathrm{E}(3)$, it is easy to have $T_g \cdot (\boldsymbol{x}(\boldsymbol{b}_{i,j}) - \boldsymbol{x}(\boldsymbol{a}_{k,j})) = O\boldsymbol{x}(\boldsymbol{b}_{i,j}) + t - O\boldsymbol{x}(\boldsymbol{a}_{k,j}) - t = O(\boldsymbol{x}(\boldsymbol{b}_{i,j}) - \boldsymbol{x}(\boldsymbol{a}_{k,j}))$. Similarly, we have $T_g \cdot (\boldsymbol{x}(\boldsymbol{b}_{i,j}) - \boldsymbol{x}_s) = O(\boldsymbol{x}(\boldsymbol{b}_{i,j}) - \boldsymbol{x}_s)$ and $T_g \cdot (\boldsymbol{x}(\boldsymbol{b}_{i,j}) - \bar{\boldsymbol{x}}(\boldsymbol{a}_k)) = O(\boldsymbol{x}(\boldsymbol{b}_{i,j}) - \bar{\boldsymbol{x}}(\boldsymbol{a}_k))$. Let $\boldsymbol{x}(\boldsymbol{b}_{i,j})'$ denote the updated pocket coordinates in Equation. 7 and 17. It is then possible to derive that the updated pocket coordinates are E(3)-equivariant as follows:

$$T_g \cdot \boldsymbol{x}(\boldsymbol{b}_{i,j}) + T_g \cdot \frac{1}{K_r} \sum_{k \in \mathcal{N}(i)} \boldsymbol{g}_{ij,kj} + T_g \cdot \frac{1}{N_l} \sum_s \boldsymbol{g}_{ij,s} \tag{19}$$

$$= O\boldsymbol{x}(\boldsymbol{b}_{i,j}) + t + O\frac{1}{K_r} \sum_{k \in \mathcal{N}(i)} \boldsymbol{g}_{ij,kj} + O\frac{1}{N_l} \sum_s \boldsymbol{g}_{ij,s} \tag{20}$$

$$= O[\boldsymbol{x}(\boldsymbol{b}_{i,j}) + \frac{1}{K_r} \sum_{k \in \mathcal{N}(i)} \boldsymbol{g}_{ij,kj} + \frac{1}{N_l} \sum_s \boldsymbol{g}_{ij,s}] + t \tag{21}$$

$$= T_g \cdot \boldsymbol{x}(\boldsymbol{b}_{i,j})'. \tag{22}$$

Similarly, for the updated ligand coordinates, we have:

$$T_g \cdot \boldsymbol{x}_s - T_g \cdot \frac{1}{|\mathcal{B}|} \sum_{i,j} \boldsymbol{g}_{ij,s} \tag{23}$$

$$= O\boldsymbol{x}_s + t - O\frac{1}{|\mathcal{B}|} \sum_{i,j} \boldsymbol{g}_{ij,s} \tag{24}$$

$$= O[\boldsymbol{x}_s - \frac{1}{|\mathcal{B}|} \sum_{i,j} \boldsymbol{g}_{ij,s}] + t \tag{25}$$

$$= T_g \cdot \boldsymbol{x}'_s. \tag{26}$$

Here, we use $\boldsymbol{x}'_s$ to denote the updated ligand coordinates in Equation. 8 and 18. Therefore, the structure refinement modules in backbone and full-atom refinement are E(3)-equivariant. □

*Proof.* The predicted residue types are obtained by applying softmax and MLPs on the hidden representations from the E(3)-invariant encoder. With the above lemmas, Theorem 1 can be proved since the E(3)-invariant encoder and the E(3)-equivariant structure refinement constitute the E(3)-equivariant generative process of FAIR. Formally, we have:

$$\{(\boldsymbol{b}_i, T_g \cdot \boldsymbol{x}(\boldsymbol{b}_i))\}_{i=1}^m = p_\theta(T_g \cdot (\mathcal{A} \setminus \mathcal{B}), T_g \cdot \mathcal{M}), \tag{27}$$

which concludes Theorem 1. □

### A.6 Huber Loss

In Equation 10, we use Huber loss [24] as the structure refinement loss for the stability of optimization, which is defined as follows:

$$l_{huber}(x, y) = \begin{cases} 0.5\,(x-y)^2, \; if \; |x-y| \; < \; \epsilon, \\ \epsilon \cdot (|x-y| \; - \; 0.5 \; \cdot \; \epsilon), \; else, \end{cases} \tag{28}$$

where $x$ and $y$ represent the predicted and ground-truth coordinates. The Huber loss has the following property: if the L1 norm of $|x-y|$ is smaller than $\epsilon$, it is MSE loss, otherwise it is L1 loss. At the beginning of the model training, the deviation between the predicted and ground-truth coordinates is large, and the L1 term makes the loss less sensitive to outliers than MSE loss. The deviation is small when the training is almost complete, and the MSE loss is near 0. In practice, we find that directly using MSE loss sometimes leads to NaN at the beginning of the training, while Huber loss makes the training procedure more stable. Following the suggestion of previous works [28, 33], we set $\epsilon = 1$ in all our experiments.

# B   Implementation Details

**PocketOptimizer** [45] [3] is a physic-based computational protein design method that predicts mutations in the binding pockets of proteins to increase affinity for a specific ligand. Its previous version is [42]. Generally, there are four main steps in PocketOptimizer: structure preparation, flexibility sampling, energy calculations, and computation of design solutions. Specifically for the energy calculations, both packing-related energies and binding-related energies are considered. As for the output design solutions, we select the top 100 designs identified by PocketOptimizer based on integer linear programming for downstream metric calculations. Following the suggestions in the original paper, we use AMBER ff14S force field [41] for energy computation and the Dunbrack rotamer library [55] for rotamer sampling for PocketOptimizer in our implementation.

**DEPACT** [8] [4] is a template-matching method that follows the two-step strategy of RosettaMatch [67] for pocket design and has better performance. It first searches the protein-ligand complexes in the database with similar ligand fragments and constructs a cluster model (a set of pocket residues). It then grafts the cluster model into the protein pocket with PACMatch, which places residues in pocket clusters on protein scaffolds by matching key atoms in the cluster model with key atoms of the protein scaffold. The qualities of the generated pockets are evaluated and ranked based on a statistical scoring function. We take the top 100 designed pockets for evaluation. The template databases are constructed in our experiments based on the corresponding training datasets for fair comparisons.

**HSRN** [28] [5] was originally developed for antibody design. It autoregressively predicts the residue sequence and docks the generated structure. To adapt HSRN to our pocket design task, we replace the antigen with the target ligand molecule to provide the context information. We finetune HSRN's hyperparameters based on its performance on the validation set. The hidden size in the message-passing network is 256, the number of layers is 4, and the number of RBF kernels is 16.

**Diffusion** [39] [6] jointly models both the sequence and structure of an antibody using diffusion probabilistic estimates and an equivariant neural network. The diffusion model first aggregates information from the antigen and the antibody during generation. It then iteratively updates each amino acid's type, position, and orientation. Since the side-chain atoms are not modeled, they are reconstructed at the last step with side-chain packing. To adapt Diffusion to our pocket design task, we replace the antigen with the target ligand molecule to provide the context information. We adjust the key hyperparameters based on the results of the validation sets. The number of layers, hidden dimension, and diffusion steps are 6, 128, and 100, respectively.

**MEAN** [33] [7] tackles antibody design as a conditional graph translation problem with the modified GMN [23] as encoders. Specifically, MEAN co-designs antibody sequence and structure via a multi-round progressive full-shot scheme, which is more efficient than auto-regressive or diffusion-based approaches. However, it only models protein backbone atoms and neglects side chains, which are important for protein-protein and protein-ligand interactions. To adapt MEAN to our pocket design task, we replace the antigen with the target ligand molecule to provide the context information. The hyperparameters are finetuned based on the validation set for better performance. We set the hidden size as 128, the number of layers as 3, and the number of iterations for decoding as 3.

**FAIR** is our full atom pocket sequence-structure co-design method. The number of layers for the atom and residue-level encoder are 6 and 2, respectively. $K_a$ and $K_r$ are set as 24 and 8 respectively. The number of attention heads is set as 4; The hidden dimension $d$ is set as 128. The standard deviation of the Gaussian noise added to the ligand coordinates in model training is 0.1.

---

[3]https://github.com/Hoecker-Lab/pocketoptimizer
[4]https://github.com/chenyaoxi/DEPACT_PACMatch
[5]https://github.com/wengong-jin/abdockgen
[6]https://github.com/luost26/diffab
[7]https://github.com/THUNLP-MT/MEAN

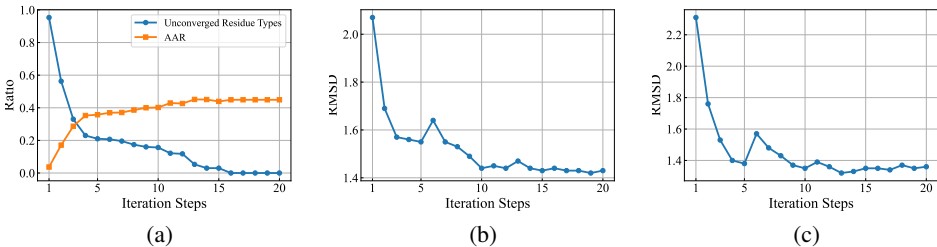

Figure 6: (a) The ratio of unconverged residue types and AAR during the iterations of FAIR on Binding MOAD; (b) The convergence plot of RMSD on CrossDocked; (c) The convergence plot of RMSD on Binding MOAD.

## C    More Experimental Results

### C.1    Pocket Generation without Backbone Reference

In FAIR, the original protein backbone coordinates are leveraged for residue-level KNN graph construction for training and generation stability. However, FAIR can also work well without protein backbone coordinates as a reference. In our supplementary experiments, FAIR can still achieve comparable performance.

### C.2    Convergence of Iterative Refinement

We show the RMSD curves in Table 6 (b) & (c). We observe the RMSD converges rapidly as the protein pocket is refined. The curve has fluctuations near step 5, potentially caused by the initialization of the side-chain atom at the beginning of full-atom refinement. The coordinates of side-chain atoms are randomly initialized near the $C_\alpha$ and may distort the backbone atoms with the structural refinement. However, the RMSD of FAIR becomes steady quickly after step 10, which again shows the effectiveness of FAIR.

### C.3    Local Geometry of the Generated Pockets

Since the RMSD used in Tables 1 and 2 reflects only the correctness of global geometry in generated protein pockets, we also calculate the RMSD of bond lengths and angles in the designed pocket to validate the local geometry following previous work [33]. We measure the average RMSD of the covalent bond lengths in the pocket backbone (C-N, C=O, and C-C). The three conventional dihedral angles in the backbone structure, i.e., $\phi, \psi, \omega$ [57] are considered, and the average RMSD of their cosine values are calculated. The results in Table 3 demonstrate that FAIR still achieves much better performance regarding the local geometry than the baseline methods.

Table 3: The average RMSD of bond lengths and dihedral angles of the generated backbone structures on the two datasets.

| Models | Bond lengths | Angles ($\phi, \psi, \omega$) |
|---|---|---|
| PocketOptimizer | 0.47 | 0.328 |
| DEPACT | 0.68 | 0.360 |
| HSRN | 0.76 | 0.437 |
| Diffusion | 0.49 | 0.344 |
| MEAN | 0.44 | 0.312 |
| FAIR | **0.35** | **0.287** |

### C.4    Hyperparameter Analysis

Here, we evaluate the influence of hyperparameter choices on FAIR performance. We focus on two hyperparameters, $T_1$ and $T_2$, for iterative refinements. In Figure 7, we observe that the performance

of FAIR is stable when $T_1$ and $T_2$ are larger than 5 and 10, respectively, indicating the refinement processes gradually converge. We set $T_1$ and $T_2$ to 5 and 10 throughout our experiments.

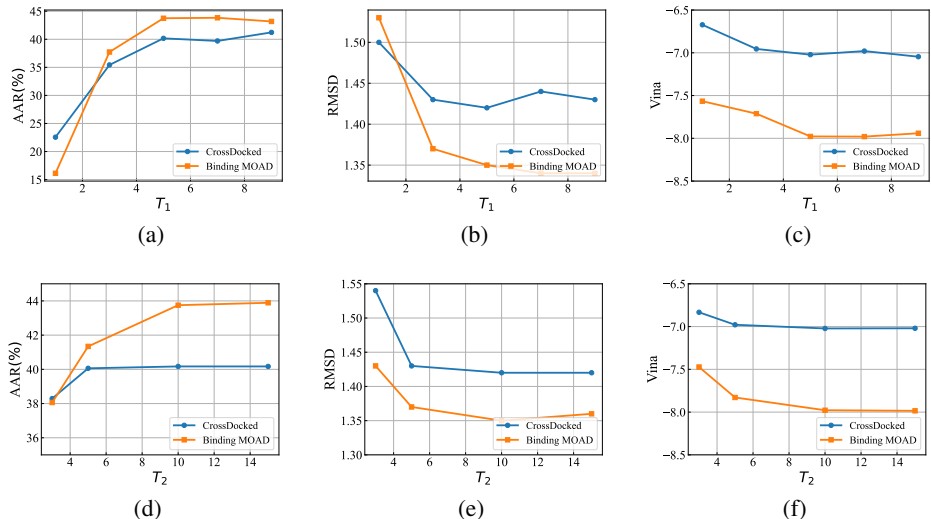

Figure 7: Hyperparameter analysis with respect to $T_1$ (first row) and $T_2$ (second row).

