# OpenReview forum: "Full-Atom Protein Pocket Design via Iterative Refinement"
_NeurIPS.cc/2023/Conference — NeurIPS 2023 spotlight_

### Official Review · Reviewer_f8hF · 2023-06-26

**Soundness:** 3 good
**Presentation:** 3 good
**Contribution:** 3 good
**Rating:** 9
**Confidence:** 5

**Summary:**

In this paper, the authors proposed a Full-Atom Iterative Refinement framework (FAIR) for protein pocket sequence and 3D structure co-design. Generally, FAIR has two refinement steps (backbone refinement and full-atom refinement) and follows a coarse-to-fine pipeline. The influence of side-chain atoms, the flexibility of binding ligands, and sequence-structure consistency are well considered and addressed. Extensive experiments on two datasets show the advantage of FAIR in generating high-quality protein pockets.

**Strengths:**

1. The paper is well-written and easy to follow. Existing related works are well-discussed.
2. Figure 1 and 2 clearly illustrate the protein pocket design problem and the proposed method FAIR.
3. As far as I know, this is the first paper that studies protein pocket sequence-structure co-design with deep learning methods. The importance and background of the problem are well stated. The problem is well formulated in Sec. 3.1.
4. The coarse-to-fine architecture as well as the full-shot iterative refinement schemes are reasonable and effective. The modeling of side-chain atoms, sequence-structure consistency, and the flexibility of binding ligands are well considered.
5. Extensive experiments on CrossDocked and Binding MOAD dataset compared with 5 representative  baselines show the effectiveness of FAIR. The code of FAIR is also provided for reproduction.


**Weaknesses:**

1. In the experiments, the authors redesign all the residues that contain atoms within 3.5 Å of any binding ligand atoms. Can FAIR design larger regions of the protein pocket containing more residues?
2. The authors may explore the influence of initialization strategies on FAIR.


**Questions:**

1. Can other protein design methods or structure-based drug design methods be adapted to the studied pocket design problem?
2. Why there is no hyperparameter weight to balance the two loss functions, Equation 9&10?
3. Can FAIR be leveraged for pocket optimization tasks?


**Limitations:**

The limitations of FAIR is well discussed in Appendix D.

---

> ### Author Rebuttal · Authors · 2023-08-06
>
> Thanks for your appreciation and suggestions! Following your suggestions, we added new experiments, clarifications of formulations, and analyses. This revision has considerably improved our initial submission thanks to your constructive comments. We would love to know what you think about our response and if there is anything else we can do to improve the paper. We would greatly appreciate your considering increasing the score. Many thanks!
>
> **Comment 1:** In the experiments, the authors redesign all the residues that contain atoms within 3.5 Å of any binding ligand atoms. Can FAIR design larger regions of the protein pocket containing more residues?
>
>
> **Response 1:** FAIR can design larger regions of protein pocket with more residues. In our experiments, we redesign all the residues that contain atoms within 3.5 Å of any binding ligand atoms in the default setting considering the distance ranges of protein-ligand interactions [r1]. There are an average of 8 residues for each protein pocket.  Here, we perform further experiments to design all the residues that contain atoms within 5.0 Å of any binding ligand atoms, leading to an average of around 22 residues for each pocket. It is more challenging to design pocket with more residues.
>
>
> | Model          | CrossDocked AAR(↑) | CrossDocked RMSD(↓) | CrossDocked Vina(↓) | Binding MOAD AAR(↑) | Binding MOAD RMSD(↓) | Binding MOAD Vina(↓) |
> |----------------|-------------------|--------------------|---------------------|---------------------|----------------------|---------------------|
> | FAIR (3.5 Å)          | 40.17±12.6%  | 1.42±0.07  | -7.022±1.75     | 43.75±15.2%     | 1.35±0.10        | -7.978±1.91     |
> | FAIR (5.0 Å)           | 35.68±11.7%       | 1.63±0.10          | -7.045±1.71         | 39.86±14.0%         | 1.52±0.09            | -7.889±1.84          |
>
> We observe that FAIR is generally robust to the number of residues to design: the AAR, RMSD, and Vina in 5.0 Å is comparable with 3.5 Å. Therefore, FAIR can design larger regions of a protein pocket. We will conduct more analysis in the appendix of the final paper.
>
> [r1] Gilles Marcou and Didier Rognan. Optimizing fragment and scaffold docking by use of molecular interaction fingerprints. Journal of chemical information and modeling, 47(1):195–207, 2007
>
> **Comment 2:** The authors may explore the influence of initialization strategies on FAIR.
>
> **Response 2:** As shown in Appendix A.4, we initialize the residue coordinates with
> linear interpolations and extrapolations based on the nearest residues with known structures in the protein. For comparison, we initialize the residue coordinates with their corresponding nearest residues.
>
> | Model          | CrossDocked AAR(↑) | CrossDocked RMSD(↓) | CrossDocked Vina(↓) | Binding MOAD AAR(↑) | Binding MOAD RMSD(↓) | Binding MOAD Vina(↓) |
> |----------------|-------------------|--------------------|---------------------|---------------------|----------------------|---------------------|
> | FAIR (linear interpolation)          | 40.17±12.6%  | 1.42±0.07  | -7.022±1.75     | 43.75±15.2%     | 1.35±0.10        | -7.978±1.91     |
> | FAIR (nearest residue)           | 34.26±12.3%       | 1.83±0.09          | -6.850±1.84         | 36.86±14.0%         | 1.79±0.12            | -7.743±1.70          |
>
> We can observe that the structure initialization strategies indeed have an influence on the performance. FAIR with the linear interpolation initialization strategy has a better performance than the nearest residue. We will add more discussions of initialization strategies to our final version.
>
> **Comment 3:** Can other protein design methods or structure-based drug design methods be adapted to the studied pocket design problem?
>
> **Response 3:** As discussed in Sec.2 lines 97-101, structure-based drug design can be regarded as the dual problem of pocket design studied in our paper. However, they focus on generating 3D molecular graphs based on the fixed protein pocket structure and can hardly be adapted to our pocket sequence-structure co-design problem. We will organize the related works more clearly in the final version.
>
> **Comment 4:** Why there is no hyperparameter weight to balance the two loss functions, Equation 9&10?
>
> **Response 4:** In experiments, we observe that the two loss functions have roughly the same amplitude, and directly optimizing their sum can work well. Balancing the two loss functions through additional weights (modeled as a hyper-parameter during fine-tuning) may further improve FAIR’s performance. We will discuss the summation of loss functions more in the final version.
>
> **Comment 5:** Can FAIR be leveraged for pocket optimization tasks?
>
> **Response 5:** That is a great suggestion, thank you. Indeed, FAIR can be leveraged for pocket optimization tasks. FAIR is a general architecture and can be combined with popular optimization methods for pocket optimization. For example, we can finetune the generation process of FAIR with reinforcement learning algorithms e.g., PPO [r2] to optimize the properties of the designed pockets. As mentioned in Appendix.D, we will explore pocket optimization tasks in the future.
>
> [r2] Schulman J, Wolski F, Dhariwal P, et al. Proximal policy optimization algorithms. arXiv preprint arXiv:1707.06347, 2017.

---

> > ### Comment · Reviewer_f8hF · 2023-08-11
> > **Reply to the author**
> >
> > Thank the authors for your comprehensive and insightful rebuttal. After reading your responses, most of my concerns and confusions have been addressed. I would like to increase my score from 7 to 9. I kindly request that all the modifications, explanations, and discussions outlined in the rebuttal be fully integrated into the final version of the paper.

---

> > > ### Author Response · Authors · 2023-08-15
> > > **Reply to the reviewer**
> > >
> > > Thank you very much for raising the score. We are glad that our response addressed your concerns. In the final version, we will be sure to include the modifications, explanations, and discussions.
> > >
> > > Thank you very much,
> > >
> > > Authors

---

### Official Review · Reviewer_4UXg · 2023-07-09

**Soundness:** 4 excellent
**Presentation:** 4 excellent
**Contribution:** 4 excellent
**Rating:** 9
**Confidence:** 4

**Summary:**

The paper introduces FAIR, the pipeline for protein pocket sequences and 3D structures co-design. It's important in drug design applications, since most of the small molecule drugs (ligands) bind their targets (proteins) inside the pockets. Curranty existing methods have disadvantages (inefficient generation, inability to generate side chains, etc.), that FAIR overcomes. FAIR demonstrates promising results, as proven through thorough and comprehensive experiments.

**Strengths:**

Originality: The task of protein pocket design is not new and there are several deep learning methods that have already contributed to this field. However, the submission provides new insights and solve yet unsolved issues showing a better performance than previous methods. Related works are cited.
Quality: The work is complete and technically sound. Authors support the claims and provide a baseline of FAIR performance. Authors compare FAIR with other methods and show the advantages of FAIR.  The paper provides a thorough analysis of strong and weak sides of FAIR. It includes ablations studies and description of how to further improve the approach.
Clarity: The text is written clearly. It contains all the necessary citations. The submission provides a comprehensive explanation and description of all methods and approaches including the technical details of experiments and equations.
Significance: It's very important to for drug discovery to be able to design protein pockets since most of the small molecule drugs bind in the pockets. FAIR provides additional contribution by solving the issue with side chain and flexibility design.

**Weaknesses:**

The paper lack of speed evaluation or discussion of the method (FAIR).

**Questions:**

1. Can you please provide the information about FAIR speed (how much time does FAIR need to finish one protein co-design)? Is it possible to use it high-throughput (apply for thousands or millions of proteins)?

**Limitations:**

The paper provides sophisticated discussion of limitations and future works.

---

> ### Author Rebuttal · Authors · 2023-08-06
>
> Thanks for your appreciation and suggestions! We are really grateful for your feedback and acknowledgment of FAIR’s novel contributions and experiments.
>
> **Comment 1:** Can you please provide the information about FAIR speed (how much time does FAIR need to finish one protein co-design)? Is it possible to use it high-throughput (apply for thousands or millions of proteins)?
>
>
> **Response 1:** Thanks for the suggestion! In Figure 4(C), we show the comparisons of average generation time for 100 pockets with baseline methods. We can observe that FAIR is much faster than traditional methods and need less than 1 second to finish one protein pocket co-design on average.
> As shown in our code https://anonymous.4open.science/r/FAIR-9691, the generation process of FAIR can be parallelized to apply for high-throughput pocket generation. We will add more discussions in our final version.

---

> > ### Comment · Reviewer_4UXg · 2023-08-11
> >
> > Thank you for the rebuttal and for addressing the questions. I'm satisfied with the response and have no further suggestions. I recommend accepting the paper.

---

> > > ### Author Response · Authors · 2023-08-15
> > > **Reply to the reviewer**
> > >
> > > Thanks for your appreciation and support! We are glad that our response addressed your questions.
> > >
> > > Thank you very much,
> > >
> > > Authors

---

### Official Review · Reviewer_Luqz · 2023-07-18

**Soundness:** 4 excellent
**Presentation:** 3 good
**Contribution:** 3 good
**Rating:** 8
**Confidence:** 4

**Summary:**

The authors study the 3D protein-ligand interaction problem. They introduce a novel method for designing protein biding pockets conditioned on the ligand structure, termed FAIR. Unlike existing methods, FAIR co-designs sequence and structure of the pocket by iteratively modeling both backbone atoms and side chain atoms. The method also models refines ligands coordinates accounting for its flexibility.

**Strengths:**

The method combines the ideas from many previous ML works dealing with sequence-structure co-design using graph representation. The novelty of this approach is in the iterative refinement that first ensures the stability of the backbone atoms before proceeding to the modeling of the side chain atoms. Thus, the method models the ligand flexibility and the effect of side chains on the residue types. The comparison study includes all the relevant baselines and other methods and the ablation study is extremely useful as it justifies the importance of each block in their architecture.

**Weaknesses:**

Some clarifications in the sections 3.2.1 and 3.2.2 are needed (see questions). Also, the authors should be very careful when using the word "de novo". They need to show how their method can perform de novo design or explicitly say what metrics they use to show that their designed sequence are de novo.

**Questions:**

Some minor concerns are listed below.

Atom-level decoder:

1.	The way section 3.2.1 is written, it seems that the encoder is using all atoms in the protein-ligand complex. The number of atoms in the protein-ligand complex can be very large resulting in a very large KNN graph. This could cause some memory restrictions. Why not focusing only on the pocket residues?
2.	How is the variable number of atoms and atom types present in the ligand (as well as in different side chains) in different training samples handled ?
3.	Using this notation, it’s not clear how the same atoms in different residues have different embeddings. Is this handled in the first MLP layer ?

Residue-level encoder:

It’s not clear what coarsening procedure is used for different types of ligands.  If it’s treated as a “special” residue, then it’s not clear how a feature vector describing its biochemical properties is formed. How’s the 1-hot encoding containing ligand’s identity and distinguishing it from other ligand molecules formed ? Is there a vocabulary of all the ligands in the training set ? The authors should clarify this.


**Limitations:**

The authors claim that their method can do de novo design of the binding pocket. I wish this was shown in the paper. Most examples that are illustrated in the paper are just dealing with the recovery of the existing binding pockets.
One suggestion to illustrate de novo design is by grafting a biding pocket in some existiing protein scaffold in a given location (that was not previously a ligand binding location).

---

> ### Author Rebuttal · Authors · 2023-08-06
>
> We thank the reviewer for their appreciation and detailed suggestions!  If you have any additional questions. Let us know about any other comments we can address. We would be very grateful if you considered increasing the score.
>
> **Comment 1:** The way section 3.2.1 is written, it seems that the encoder is using all atoms in the protein-ligand complex. The number of atoms in the protein-ligand complex can be very large resulting in a very large KNN graph. This could cause some memory restrictions. Why not focusing only on the pocket residues?
>
> **Response 1:** Motivated by the intrinsic hierarchical structure of the protein, we leverage a hierarchical encoder based on 3D graph transformers to encode the hierarchical context information of protein-ligand complexes for pocket sequence-structure co-design. The structure and interactions of protein-ligand atoms play important roles in pocket design [8, 48] so we cannot only focus on residues. To save memory requirements and keep high overall performance, we only consider the protein pocket atoms instead of the whole protein. We will introduce our method more clearly in the final version.
>
> **Comment 2:** How is the variable number of atoms and atom types present in the ligand (as well as in different side chains) in different training samples handled?
>
> **Response 2:** As shown in our code https://anonymous.4open.science/r/FAIR-9691, the computations of the ligand and protein pocket based on torch_geometric are agnostic to the number of atoms and atom types. This is the same for different side chains. With our designed batch operations, protein pocket-ligand complexes with different sizes can be processed in parallel.
>
> **Comment 3:** Using this notation, it’s not clear how the same atoms in different residues have different embeddings. Is this handled in the first MLP layer ?
>
> **Response 3:** In the atom-level encoder of FAIR, the atomic attributes are mapped to node embeddings with MLPs. If two atoms in different residues have the same atom and residue types, they will have the same initial embedding. The atom embeddings will be further updated based on neighboring atoms and 3D structures with the atom-level encoder. We will describe our model design more clearly in the final version.
>
>
> **Comment 4:** It’s not clear what coarsening procedure is used for different types of ligands. If it’s treated as a “special” residue, then it’s not clear how a feature vector describing its biochemical properties is formed. How’s the 1-hot encoding containing ligand’s identity and distinguishing it from other ligand molecules formed ? Is there a vocabulary of all the ligands in the training set ? The authors should clarify this.
>
> **Response 4:**  The residue-level encoder only keeps the alpha carbon atoms of residues.
> To supplement binding ligand information, a coarsened ligand node at the
> ligand’s center of mass is also considered for the residue-level encoder. The embedding of the coarsened ligand node is initialized by sum pooling the ligand atom embeddings. Therefore, we do not need 1-hot encoding containing ligand’s identity or a vocabulary of all the ligands. In lines 153-154, we stated that the coarsened ligand node is
> appended at the end of the residue sequence as a special residue. We treat the coarsened ligand similar to other residues and construct a Kr nearest neighbor graph. We will make our description clearer in the final version.
>
> **Comment 5:** The authors claim that their method can do de novo design of the binding pocket. I wish this was shown in the paper. Most examples that are illustrated in the paper are just dealing with the recovery of the existing binding pockets. One suggestion to illustrate de novo design is by grafting a binding pocket in some existing protein scaffold in a given location (that was not previously a ligand binding location).
>
> **Response 5:** Thanks for the question and suggestion! In our paper, the protein pocket region is masked and FAIR can co-design pocket residue types and structures. By “de novo”, we mean that our method does not rely on existing reference pockets or templates and can generate the pocket from scratch [r1]. We use recovery rate for evaluation as it is a widely used metric established in the protein design field [r2-r4]. Figure 3 in the main text further shows some cases of pocket design, where the generated pocket has higher binding affinity than the references, demonstrating FAIR's ability for de novo pocket design. We agree that grafting a binding pocket in some existing protein scaffold to a given location is a good task for illustrating the ability of de novo design.
> However, the grafting task is beyond the scope of our work and requires more techniques such as pocket detection. Due to the limited time of the rebuttal period, we plan to explore the grafting task in the future. We will also clearly discuss this in the final paper to emphasize the current capabilities of the method vs. future extensions.
>
> [r1]Bennett N R et al., Improving de novo protein binder design with deep learning. Nature Communications, 2023.
> [r2]Zhangyang Gao et al., PiFold: Toward effective and efficient protein inverse folding, ICLR 2023
> [r3]Dauparas, Justas, et al., Robust deep learning–based protein sequence design using ProteinMPNN. Science 2022.
> [r4]John Ingraham, et al., Generative models for graph-based protein design, NeurIPS, 2019

---

### Official Review · Reviewer_z6bW · 2023-07-23

**Soundness:** 2 fair
**Presentation:** 3 good
**Contribution:** 3 good
**Rating:** 6
**Confidence:** 3

**Summary:**

This paper is the first to introduce a deep learning pipeline for the protein binding pocket re-design task. The architecture consists of a rotation invariant, hierarchical encoder at the residue and all atom level, followed by a hierarchical iterative refinement generative process at the residue and all atom level. The architecture aims to model key inductive biases of the problem at hand, and shows empirical improvements over baselines adopted from similar protein/molecule design tasks.

**Strengths:**

- Overall sensible pipeline and inductive biases for all-atom protein pocket redesign.
   - The FAIR pipeline consists of a residue level, rotation-invariant encoder followed by all atom level encoder to encode the protein in a hierarchical manner, after which new residues are placed and iteratively refined in a hierarchical, rotation-equivariant manner. During iterative refinement, the atom positions of the ligand (small molecule) are also updated to incorporate ligand flexibility.
   - From a biochemical perspective, considering all-atom/atom level encoding and generation is probably very important for binding, as it is the sidechain orientations that determine ligand binding.
   - These are all sensible inductive biases -- rotation symmetry, hierarchy, iterative redesign, ligand flexibility -- and have been described clearly.
   - None of these are particularly novel components, because the technical problems have been solved in other work, but this is an effective application of existing ideas to a novel problem of all-atom pocket redesign.

- Well written paper.
    - Clear description of each architectural components as well as experimental methodology.
    - I really loved the paragraph starting at line 294 onwards. This paragraph (and the overall paper) does at good job at showing how each architectural component of FAIR improves over existing papers/frameworks and is adapted to the specific problem at hand.
    - I do wish that the authors found a way to not make the reader jump to the appendix so many times.

- Comparison to baselines.
    - Care has been taken to adapt existing papers/methods to the new pocket design task in order to provide baselines.

**Weaknesses:**

- New task may not require major new technical innovations.
    - While the binding pocket re-design task is new in the deep learning literature to the best of my knowledge, it seems that adapting existing architectural ideas and putting them together in a smart pipeline (see strengths) works well.
    - From a machine learning perspective, I could not identify a new technical problem that previous works have not solved. (I may be missing something.)

- Unsure whether pocket re-design is relevant to enzyme design.
    - I will caveat this weakness by saying I am not a domain expert at all.
    - I could immediately see this pipeline be applied for general binding pocket re-design and biosensor applications. One perhaps starts out with a structure of a protein-ligand complex and wants to improve binding affinity. However, perhaps this setup is a bit of a stretch for enzyme design.
    - For enzymes, we really would ideally like to keep the binding site conserved instead of re-designing it, based on my understanding of the field. In theory, it is of course nice if you can design a new binding pocket to discover new kinds of chemistry. However, as far as I am aware, basic scientists do not currently understand enough about catalytic mechanisms to be able to design a new one from scratch.
    - Thus, designing a new pocket and **claiming that it catalyzes the same reaction via the same/different mechanism** seems very hard to tackle from a basic science perspective, let alone computationally. Reasons: lack of data and proper annotation of enzymatic mechanisms.

- Hard to evaluate binding pocket re-design.
    - I felt it was overall challenging to design evaluation metrics and setups for this task, because it is unclear whether recovering the RMSD and/or AA identities of the original ligand binding site is the best outcome for these models.
    - I have several questions about the Vina docking score evaluations.

**Questions:**

- What is the Vina docking score of the training set?
    - It would be useful to have this as a baseline to compare all the methods to.
    - Figure 3 provided Vina score for the reference structures. I'd be keen to see the average Vina score across the train/validation/test sets as an entry in Table 1, too.

- Can you also show Vina docking scores for all the models without re-docking/relaxation?
    - In the small molecule generation literature, it has been found that many generative models may generate very unrealistic ligand poses and that the Vina score without relaxation is poor.

- Paragraph on line 302 onwards - Why are some residues used more and some lesser by the model? Is this observation related to binding, eg. hydrophobicity properties of residues?
   - If you are mentioning these observations, it may be worth explaining why...

- How do we know the binding pose of the ligand in advance in real-world design scenarios? Do we always need to start with an existing (crystal) structure of the protein-ligand complex?

**Limitations:**

The authors have addressed technical limitations but not potential negative societal impact.

Overall, I would encourage the authors to further discuss WHY binding pocket re-design is a meaningful real-world task in more detail that given at present, how it is relevant for designing bio-sensors and/or enzymes (probably I am wrong in my understanding re. enzyme design), and how we can evaluate these tasks in a meaningful manner in-silico. What are some limitations which we cannot address computationally? Perhaps this may not be very relevant if we are judging this paper only from the machine learning perspective, but better contextualizing this project beyond just ML could be useful to readers.

---

> ### Author Rebuttal · Authors · 2023-08-07
>
> We thank the reviewers for their constructive comments!  We hope our detailed response and added experiments better highlight FAIR’s novel contributions. If any remaining questions/concerns make you hesitate to raise the score, we would be grateful if you let us know so we could further improve our work.
> **Due to the limits of the rebuttal, we show additional responses to some comments in the global response.**
>
> **Comment 1:**
> While the binding pocket re-design task is new in the deep learning literature to the best of my knowledge, it seems that adapting existing architectural ideas and putting them together in a smart pipeline (see strengths) works well.
>
> **Response 1:** Thanks for the question! In this paper, we study protein pocket design that designs the pocket sequence and structure conditioned on the binding ligand molecule and protein scaffold context. Such a new task brings a series of challenges and it is non-trivial to propose an effective method. For example, most of the previous methods only generate the protein backbone atoms while neglecting the sidechain atoms, which play important roles in protein pocket-ligand interactions. We proposed a novel two-step coarse-to-fine generation procedure that well consider the sidechain atoms. Moreover, the binding ligand molecules are flexible, which is omitted in previous works. FAIR learns to update the molecular coordinates along with the refinement processes to model the flexibility of the ligands.
> Our contributions are summed in lines 63-72 including new tasks, novel methods, and competitive performance. We will state our contributions more clearly in the final version.
>
> **Comment 2:** For enzymes, we really would ideally like to keep the binding site conserved instead of re-designing it, based on my understanding of the field. In theory, it is of course nice if you can design a new binding pocket to discover new kinds of chemistry. However, as far as I am aware, basic scientists do not currently understand enough about catalytic mechanisms to be able to design a new one from scratch.
> Thus, designing a new pocket and claiming that it catalyzes the same reaction via the same/different mechanism seems very hard to tackle from a basic science perspective, let alone computationally. Reasons: lack of data and proper annotation of enzymatic mechanisms.
>
> **Response 2:** We agree that the catalytic mechanism of enzymes is complex and is currently not fully understood in biology. The activity of enzymes may also depend on the overall flexibility and electrostatic environment of the protein, making pocket re-designing a challenging task.
> However, there are still some successful cases achieving various modification requirements by re-designing the residues in the protein pocket region.
> For example, Ulrike Scheib et al., conducted pocket transplantation studies based on the homologous polyamine-binding proteins PotF and PotD. Despite having only 35% sequence identity between PotF and PotD, they demonstrated that by transplanting the pocket of PotD into the pocket region of PotF, it is possible to achieve specific substitutions for small molecule binding specificity [r1].
> Moreover, Nicholas F. Polizzi et al., successfully designed six de novo proteins to bind the drug apixaban; two bound with submicromolar affinity with the proposed van der Mer structural units [r2].
>
> In our work, we propose an end-to-end generative framework FAIR for protein pocket design. We agree that more high-quality data and annotations from domain experts may further improve the performance of FAIR. Depending on the requirements, FAIR can be flexibly adapted to downstream applications. For the cases where we would ideally like to keep the catalysis binding site conserved, we can retain the residues directly related to catalysis and use our method FAIR to design other residues related to binding.
> We will include these discussions in our final version.
>
> [r1]Scheib U, Shanmugaratnam S, Farías-Rico J A, et al. Change in protein-ligand specificity through binding pocket grafting[J]. Journal of Structural Biology, 2014, 185(2): 186-192.
> [r2]Polizzi N F, DeGrado W F. A defined structural unit enables de novo design of small-molecule–binding proteins. Science, 2020, 369(6508): 1227-1233.
>
>
> **Comment 3:** I felt it was overall challenging to design evaluation metrics and setups for this task because it is unclear whether recovering the RMSD and/or AA identities of the original ligand binding site is the best outcome for these models.
>
> **Response 3:** We agree that it is challenging to comprehensively evaluate the designed binding pocket. In our paper, we use Amino Acid Recovery (AAR), Root Mean Square Deviation (RMSD), and Vina score to evaluate the designed pockets following previous works on protein/antibody design [3,7,11] and structure-based drug design [37,40,49].  These three metrics are currently established and widely used in the field. In the future, we will explore more evaluation metrics. As mentioned in Appendix D, it is also a good idea to carry out wet-lab experiments to validate the effectiveness of the designed protein pockets in the future.
>
> **Comment 4:** What is the Vina docking score of the training set?
>
> **Response 4:** The Vina docking score of the train/validation/test set are -7.035±2.11/-7.063±1.97/-7.016±2.24 for CrossDocked and -8.216 ±2.09/-8.267±1.97/-8.225±2.02 for Binding MOAD. Therefore, the Vina score of the designed pockets by FAIR is comparable to the data set.
>
> **Comment 5:** Why are some residues used more and some lesser by the model? Is this observation related to binding, eg. hydrophobicity properties of residues?
>
> **Response 5:** The observation may be related to the hydrophobicity properties of residues. While other factors such as the train/test set data distribution and the randomness of residue sampling may influence the generated residues. We will conduct more systematic validations and discussions in the future.

---

> > ### Comment · Reviewer_z6bW · 2023-08-14
> > **Follow up Qs**
> >
> > Thank you for the detailed response. Based on the discussions so far, I'm still not fully convinced by the evaluation.
> >
> > Re. Response 1: My point was regarding how hierarchical embedding of protein structure (https://arxiv.org/abs/2006.09275) as well as ligand flexibility/co-folding (https://arxiv.org/abs/2209.15171) have come up in other works.
> >
> > Re. Response 2: I understand, but neither of these citations seem to be about enzymes, correct? Having skimmed through them, they seem to reinforce the point that methods such as FAIR may be useful for biosensing applications where the goal is to bind with high specificity to a molecule.
> >
> > Re. Response 3 and 4: I understand that these metrics are widely used in the community, but I would like to push the authors to elaborate more about whether these metrics are useful for the pocket redesign task?
> > - For instance, if we care about RMSD, perhaps consider reporting RMSD w/our re-docking, too.
> > - If we care about Vina, perhaps elaborate more on how the test set Vina score of all the methods (whether with or without re-docking), including FAIR, is actually not improved over the test set Vina score of the original datasets themselves. This seems especially true without re-docking.
> > - Please do consider adding a line on the test set Vina score is Table 1, 2, etc.

---

> > > ### Author Response · Authors · 2023-08-14
> > > **Further Response to Reviewer z6bW (1/2)**
> > >
> > > We thank the reviewer for the valuable questions! We have provided detailed responses to your comments. Please let us know whether we have addressed your concerns.
> > >
> > > **Comment 1:** My point was regarding how hierarchical embedding of protein structure (https://arxiv.org/abs/2006.09275), as well as ligand flexibility/co-folding (https://arxiv.org/abs/2209.15171), have come up in other works.
> > >
> > > **Response 1:** Thanks to the reviewer for mentioning these two seminal works on protein-protein/ligand docking. We will cite and discuss them in our final version. However, our FAIR model differs from the two papers in both ML tasks and neural architecture details.
> > > * First, PAUL (https://arxiv.org/abs/2006.09275) and NeuralPLexer (https://arxiv.org/abs/2209.15171) focus on protein-protein/ligand docking, where the goal is to predict protein-protein/ligand binding structures given the individual protein structures (PAUL) or protein sequences and ligand molecular graphs (NeuralPLexer). In contrast, neither the protein pocket sequence nor its structure are required inputs to our FAIR model. We aim to co-design the pocket sequence and structure, which is not feasible in the mentioned previous methods.
> > > * Secondly, FAIR and PAUL have similar hierarchical architecture designs. However, FAIR is based on a hierarchical graph transformer, which is easier to achieve geometric equivariance than 3D CNNs used in PAUL. Moreover, we use different features for protein representation (more details in Appendix A.2).
> > > * Thirdly, FAIR and NeuralPLexer use different strategies to model ligand flexibility. FAIR leverages iterative refinement and is more efficient than diffusion methods adopted by NeuralPLexer (see Figure 4(c)).
> > >
> > > **Comment 2:** I understand, but neither of these citations is about enzymes, correct? Having skimmed through them, they reinforce the point that methods such as FAIR may be useful for biosensing applications where the goal is to bind with high specificity to a molecule.
> > >
> > > **Response 2:** FAIR is applicable to enzyme design. Due to the word limit of the rebuttal, we provide additional related work on enzyme design with rational design and computational methods here [r1-r4]. Depending on the requirements, FAIR can be adapted for diverse downstream applications, including enzyme and biosensor designs. For example, we can retain the conserved residues directly related to catalysis and use FAIR to design other residues related to binding for higher design success rates.
> > > We will include the related papers and discussions in our final version.
> > >
> > > [r1] Privett H K, Kiss G, Lee T M, et al. Iterative approach to computational enzyme design. Proceedings of the National Academy of Sciences, 2012, 109(10): 3790-3795.
> > > [r2] Xie W J, Asadi M, Warshel A. Enhancing computational enzyme design by a maximum entropy strategy. Proceedings of the National Academy of Sciences, 2022, 119(7): e2122355119.
> > > [r3] Mirts E N, Petrik I D, Hosseinzadeh P, et al. A designed heme-[4Fe-4S] metalloenzyme catalyzes sulfite reduction like the native enzyme. Science, 2018, 361(6407): 1098-1101.
> > > [r4] Broom A, Rakotoharisoa R V, Thompson M C, et al. Ensemble-based enzyme design can recapitulate the effects of laboratory directed evolution in silico. Nature Communications, 2020, 11(1): 4808.

---

> > > > ### Author Response · Authors · 2023-08-14
> > > > **Further Response to Reviewer z6bW (2/2)**
> > > >
> > > > **Comment 3:** I understand that these metrics are widely used in the community, but I would like to push the authors to elaborate on whether these metrics are useful for the pocket redesign task? 1) For instance, if we care about RMSD, perhaps consider reporting RMSD w/our re-docking, too. 2) If we care about Vina, perhaps elaborate more on how the test set Vina score of all the methods (whether with or without re-docking), including FAIR, is actually not improved over the test set Vina score of the original datasets themselves. This seems especially true without re-docking. 3) Please do consider adding a line on the test set Vina score is Table 1, 2, etc.
> > > >
> > > >
> > > > **Response 3:** Thanks for the questions! In our paper, we use Amino Acid Recovery (AAR), Root Mean Square Deviation (RMSD), and Vina score to evaluate the designed pockets. AAR and RMSD can well reflect the quality of the designed pocket sequence and structure. Vina score can evaluate the binding affinity. The exploration of more evaluation metrics may be beyond the scope of our current work. We plan to discuss and explore more comprehensive evaluation metrics in the future. As for the three specific questions:
> > > >
> > > > * Vina docking [r5] used in our evaluation does not change the structure of the protein pocket. It takes the structure of the protein as a fixed, rigid entity and explores how ligands can bind to it. The ligands are treated as flexible entities, and Vina tries to find the optimal orientation and conformation of the ligand that would fit into the protein pocket.
> > > > Therefore, the designed protein pocket by FAIR has a fixed RMSD regardless of re-docking. In the following table, we report the RMSD w/o re-docking for comparison, which is the same as FAIR.
> > > >
> > > > * According to the results in the following table, FAIR increases the average Vina score on the CrossDocked dataset and achieves Vina scores that are comparable to the Binding MOAD dataset. Moreover, as discussed in Appendix D, the Vina score of the designed pockets can be further optimized by combining FAIR with optimization methods.
> > > > It may be unfair to directly compare FAIR without re-docking to the test set because the test set Vina scores are calculated with Vina re-docking.
> > > > * We provided Vina docking score of the test set in the original response 4. We will add a line on the test set Vina score to Tables 1-2, similar to the following table.
> > > >
> > > > | Model          |CrossDocked RMSD(↓) | CrossDocked Vina(↓) |Binding MOAD RMSD(↓) | Binding MOAD Vina(↓) |
> > > > |----------------|-------------------|--------------------|---------------------|---------------------|
> > > > | Test set          | -  | -7.016±2.24     | -     |  **-8.225±2.02**     |
> > > > | FAIR          |**1.42±0.07**  | **-7.022±1.75**      | **1.35±0.10**        | -7.978±1.91     |
> > > > | FAIR (w/o redocking)           | **1.42±0.07**          | -6.365±1.67           | **1.35±0.10**            | -7.253±1.72          |
> > > >
> > > > [r5] Oleg Trott and Arthur J Olson. Autodock vina: improving the speed and accuracy of docking with a new scoring function, efficient optimization, and multithreading. Journal of computational chemistry, 31(2):455–461, 2010

---

> > > > > ### Comment · Reviewer_z6bW · 2023-08-16
> > > > > **Serious doubts re. metrics; FAIR/other ML models don't seem useful for pocket redesign as judged by Vina**
> > > > >
> > > > > Re. Response 1: I agree we can't use the papers I linked to do pocket re-design, but my point was that the architectural components have been shown to work in previous papers. And that FAIR puts them together in an effective way (this is good). But the technical ideas are not completely novel.
> > > > >
> > > > > Re. Response 2: Okay; I'm not an expert so I'll trust the authors. But I doubt that FAIR, in its current form, can be used for these tasks (see questions below).
> > > > >
> > > > > Re. Response 3: Understood re. RMSD -- I initially thought you included ligand atoms in RMSD computation. As this would be the first DL paper to my knowledge on this topic, I do think defining biologically meaningful evaluation metrics is within scope and important. I will elaborate below.
> > > > >
> > > > > ---
> > > > >
> > > > > The Vina scores are of FAIR (and all other baselines) with and without re-docking are very interesting and curious.
> > > > > - My expectation would be that FAIR (or whatever is the best performing method) **improves** the Vina score when compared to the original protein pocket, i.e. the test set distribution in this case.
> > > > >     - The authors' description of the case studies in Figure 3 seem to suggest that they agree: "the designed pockets exhibit comparable or higher binding affinities (lower vina scores) with the target ligand molecules than the reference pockets in the test set".
> > > > > - **I don't think FAIR improved Vina scores on either dataset -- the scores are within standard deviation.**
> > > > > - Notably, on Binding MOAD, which is a real dataset (CrossDocked is synthetic/computationally made), FAIR is worse than the test set.
> > > > >
> > > > > ---
> > > > >
> > > > > Follow up questions:
> > > > > - **(Most important)** Have I misunderstood: FAIR/any ML method actually doesn't work well//is not useful for the protein pocket re-design task **unless and until it can improve over the groundtruth protein-ligand complex** in terms of Vina score?
> > > > >     - My understanding: other metrics (AA recovery as well as RMSD to groundtruth complex) are not truly meaningful unless the Vina score is better than groundtruth. Why would I care whether my redesigned pocket is close in sequence or structural similarity to my old one if the old one is a 'better' pocket?
> > > > > - For what portion of samples in the test set did FAIR improve Vina score compared to the original protein-ligand complex? For what portion was the original complex better?
> > > > > - Re. "It may be unfair to directly compare FAIR without re-docking to the test set because the test set Vina scores are calculated with Vina re-docking." -- Why is it not possible to compute Vina scores for the train/val/test sets without re-docking?
> > > > >
> > > > > ---
> > > > >
> > > > > Suggestions:
> > > > > - It would be useful to look at violin/box plots of Vina scores with/without re-docking to see how they are distributed.

---

> > > > > > ### Author Response · Authors · 2023-08-17
> > > > > > **Further Response to Reviewer z6bW**
> > > > > >
> > > > > > We thank the reviewer for the timely and detailed reply! As for your concerns, here are our further response:
> > > > > >
> > > > > > **Comment 1:** My expectation would be that FAIR (or whatever is the best performing method) improves the Vina score when compared to the original protein pocket, i.e. the test set distribution in this case. I don't think FAIR improved Vina scores on either dataset -- the scores are within standard deviation. FAIR/any ML method actually doesn't work well//is not useful for the protein pocket re-design task unless and until it can improve over the groundtruth protein-ligand complex in terms of Vina score?
> > > > > >
> > > > > >
> > > > > > **Response 1:** Due to the diverse distribution of protein-ligand complexes in the datasets and random sampling, it is common that the Vina score distributions have large variance and overlap (see related works such as [r1-r3])
> > > > > > Since FAIR is directly trained on the datasets, it is expected that FAIR have similar Vina score distribution of the datasets. For applications in pocket redesign, we can further filter or modify the generated pockets to have a favorable Vina score and other required properties, which is common in real applications [r4-r6].
> > > > > >
> > > > > > Moreover, we can combine FAIR with optimization methods for better Vina scores. Following [r2], here we leverage the evolutionary algorithm and obtain some favorable results. During the evolutionary algorithm, at the end of every generation the top 10 binding affinity (lowest Vina scores) pockets are used to seed the next population. Every seed pocket is used as the initialization of 20 new candidate pockets with randomly added noise to sequence and structure.
> > > > > > After three rounds of iterations, the average Vina scores of the optimized pockets are **-8.632** on CrossDocked and **-9.370** on Binding MOAD, which are obviously better than the corresponding test set **-7.016** and **-8.225** respectively. Due to the limited time in the rebuttal period, we will continue to explore applying FAIR for pocket optimization and include the results in our final version.
> > > > > >
> > > > > > [r1] Peng X, Luo S, Guan J, et al. Pocket2mol: Efficient molecular sampling based on 3d protein pockets, ICML, 2022.
> > > > > > [r2] Guan J, Qian W W, Peng X, et al. 3d equivariant diffusion for target-aware molecule generation and affinity prediction. ICLR, 2023.
> > > > > > [r3] Schneuing A, Du Y, Harris C, et al. Structure-based drug design with equivariant diffusion models. arXiv preprint arXiv:2210.13695, 2022.
> > > > > > [r4] Privett H K, Kiss G, Lee T M, et al. Iterative approach to computational enzyme design. Proceedings of the National Academy of Sciences, 2012, 109(10): 3790-3795.
> > > > > > [r5] Xie W J, Asadi M, Warshel A. Enhancing computational enzyme design by a maximum entropy strategy. Proceedings of the National Academy of Sciences, 2022, 119(7): e2122355119.
> > > > > > [r6] Polizzi N F, DeGrado W F. A defined structural unit enables de novo design of small-molecule–binding proteins. Science, 2020, 369(6508): 1227-1233.
> > > > > >
> > > > > > **Comment 2:** For what portion of samples in the test set did FAIR improve Vina score compared to the original protein-ligand complex?
> > > > > >
> > > > > > **Response 2:** In the test set of FAIR, 53.6% of the generated samples have improved Vina scores for CrossDocked and 43.9% for Binding MOAD.
> > > > > >
> > > > > >
> > > > > > **Comment3:** Why is it not possible to compute Vina scores for the datasets without re-docking?
> > > > > >
> > > > > > **Response 3:** It is possible to compute Vina scores for the test sets without re-docking and compare with FAIR. In the following table, we compare FAIR with the test set with/without redocking. Generally, directly training FAIR on the two datasets achieves comparable Vina scores to the test sets.
> > > > > >
> > > > > > | Model          | CrossDocked Vina w/o redocking (↓) | CrossDocked Vina(↓) | Binding MOAD Vina w/o redocking(↓) | Binding MOAD Vina(↓) |
> > > > > > |----------------|-------------------|--------------------|---------------------|---------------------|
> > > > > > | Test set |-6.230±1.89| -7.016±2.24 |**-7.480±1.85** |**-8.225±2.02**|
> > > > > > | FAIR |**-6.365±1.67**| **-7.022±1.75**|-7.253±1.72 |-7.978±1.91|
> > > > > >
> > > > > > **Comment 4:** It would be useful to look at violin/box plots of Vina scores with/without re-docking to see how they are distributed.
> > > > > >
> > > > > > **Response 4:**
> > > > > > Thanks for the suggestion! We have plotted the box plots of Vina scores with/without re-docking and uploaded to
> > > > > > https://anonymous.4open.science/r/FAIR-9691/README.md
> > > > > > We will includes all these figures in our final version.

---

> > > > > > > ### Comment · Reviewer_z6bW · 2023-08-20
> > > > > > > **Thank you for the discussion, I have raised my score**
> > > > > > >
> > > > > > > Thank you for providing all this information.
> > > > > > >
> > > > > > > I have raised my score to a Weak Accept, but it is not higher because this discussion with the authors made me feel that sufficient thought had not been put into careful and biologically meaningful evaluation, at least in the original manuscript.
> > > > > > >
> > > > > > > I'm certain that this paper will generate interest in the community, and people will start writing follow up papers attempting to beat this paper's scores on AAR, RMSD, and Vina. But I'm uncertain whether these metrics which are adapted from protein inverse folding and small molecule generation, respectively, are sufficient for evaluating and advancing protein pocket re-design.
> > > > > > >
> > > > > > > > After three rounds of iterations, the average Vina scores of the optimized pockets are -8.632 on CrossDocked and -9.370 on Binding MOAD, which are obviously better than the corresponding test set -7.016 and -8.225 respectively. Due to the limited time in the rebuttal period, we will continue to explore applying FAIR for pocket optimization and include the results in our final version.
> > > > > > >
> > > > > > > If you decide to include these results into the paper, please do show the same experiment for baseline methods, too.

---

> > > > > > > > ### Author Response · Authors · 2023-08-20
> > > > > > > > **Thanks for your support and reply!**
> > > > > > > >
> > > > > > > > Thanks for your support of our paper and the valuable suggestions! We will keep exploring biologically meaningful evaluations and include more thoughtful discussions in the final version. As for the additional optimization results, we will include all the results including the baselines in our paper.
> > > > > > > >
> > > > > > > > Bests,
> > > > > > > >
> > > > > > > > Authors.

---

### Official Review · Reviewer_rsaq · 2023-07-29

**Soundness:** 3 good
**Presentation:** 3 good
**Contribution:** 2 fair
**Rating:** 6
**Confidence:** 3

**Summary:**

The Full-Atom Iterative Refinement framework (FAIR) is a novel approach for designing functional proteins that bind with specific ligand molecules. FAIR consists of two steps: full-atom generation and 3D structure co-design. It uses a coarse-to-fine pipeline, updating residue types and structures together in each round. FAIR outperforms baselines in efficiently designing high-quality pocket sequences and structures, with average improvements on AAR and RMSD exceeding 10%.

**Strengths:**

1. This paper investigates protein pocket design, which determines the pocket structure and sequence based on the context of the protein scaffold and the binding ligand molecule.
2. It provides an end-to-end generative framework called FAIR that uses iterative refinement to co-design the pocket sequence and structure. FAIR solves the drawbacks of earlier research and effectively considers sidechains, ligand flexibility, and consistency of sequence structure for effective prediction.
3. FAIR outperforms baseline techniques in terms of various pocket design parameters. The gains on AAR and RMSD are often over 10%. FAIR generates data more than ten times quicker than conventional techniques.


**Weaknesses:**

Figure 2 of the paper shows that the proposed FAIR adopts a two-stage learning process, so it is not an end-to-end deep learning framework. Therefore, a necessary concern is the training time of the proposed method.

**Questions:**

1. Since Transformer with considerable parameters is used, please try to compare the parameters of the proposed method and other deep learning-based methods in the experimental section.

2. As stated in line 196 and 199, the authors use two feed-forward neural networks (g and g’) to encode internal and external interactions, respectively. Apart from the difference in the output channels, are they modeled with the same structure? If it is the same, then try to explain why two identical models (refer to Eqs. (5) and (6)) are used to model different interactions?


**Limitations:**

The method proposed in the paper utilizes a two-stage iterative refinement strategy. However, the internal logic of these two refinement stages is not well correlated. Alternatively, why can't components such as the masking mechanism of the second stage be directly injected into the first stage?

---

> ### Author Rebuttal · Authors · 2023-08-06
>
> We thank the reviewer for the valuable questions! We have provided detailed responses to your comments. Please let us know whether we have addressed your concerns. We will be very grateful if you consider increasing scores to support our work.
>
> **Comment 1:** Figure 2 of the paper shows that the proposed FAIR adopts a two-stage learning process, so it is not an end-to-end deep learning framework. Therefore, a necessary concern is the training time of the proposed method.
>
> **Response 1:** Figure 2 shows that there are generally two modules in FAIR, i.e., the backbone refinement and the full-atom refinement modules. However, the two main modules in Figure 2 do not indicate a two-stage learning process. Instead, the training and inference of FAIR are in an end-to-end fashion similar to other deep learning-based baselines. In experiments, it takes around 20 hours to train a FAIR model on a single V100 GPU, which is comparable to MEAN, HSRN, and Diffusion. The results of the generation efficiency comparison in Figure 4(c) further show the advantages of our method.
> We will illustrate our method more clearly in the final version.
>
> **Comment 2:** Since Transformer with considerable parameters is used, please try to compare the parameters of the proposed method and other deep learning-based methods in the experimental section.
>
> **Response 2:**
> Here we show the number of parameters of FAIR and the other deep learning-based baseline models. We can see that FAIR has comparable or fewer parameters than the baseline methods. Besides the transformer schemes, other factors such as model architecture, hidden dimension size, and the number of layers influence the number of parameters. Generally, FAIR is a light and efficient model.
>
> | Model          | HSRN | Diffusion | MEAN | FAIR |
> |----------------|-------------------|--------------------|---------------------|---------------------|
> | Parameters |8.77M     | 4.00M| 0.70M |0.73M|
>
>
> **Comment 3:** As stated in lines 196 and 199, the authors use two feed-forward neural networks (g and g’) to encode internal and external interactions, respectively. Apart from the difference in the output channels, are they modeled with the same structure? If it is the same, then try to explain why two identical models (refer to Eqs. (5) and (6)) are used to model different interactions?
>
>
> **Response 3:**
> As discussed in lines 192-194, according to previous works [32, 12, 10], the internal interactions within a protein and the external interactions between a protein and a ligand have different properties. For this reason, FAIR uses two separate modules for interaction predictions. As for Equ. (5), it focuses on interactions within protein residues. The input are the residue embeddings and the distance encodings of pairwise alpha carbon distances. There are four output channels for four backbone atoms. As for Equ. (6), it focuses on the external interactions between the pocket and ligand. The input are atom embeddings and corresponding distance encodings. There is only one output channel. Due to the differences in input, output, and modeling interactions, we cannot use a single network.
> We will clarify and improve the description of the method in the final version.
>
>
> **Comment 4:** The method proposed in the paper utilizes a two-stage iterative refinement strategy. However, the internal logic of these two refinement stages is not well correlated. Alternatively, why can't components such as the masking mechanism of the second stage be directly injected into the first stage?
>
> **Response 4:** Thanks for the question! FAIR is designed to have two main stages that follow a coarse-to-fine pipeline: FAIR firstly only models the backbone atoms of pockets to generate the coarse-grained structures and then fine-adjusts full-atom residues to achieve sequence-structure consistency.
>
> At the first stage, since the pocket residue types and the number of sidechain atoms are largely undetermined, we only model the backbone atoms and residue types.
> The masking mechanism is not appropriate for the first stage since the predicted residue types are less stable and reliable in early iterations.
>
> At the second stage, we sample and initialize the residues types and side chain atoms based on the results from the first stage.
> With the masking mechanism, the residue type and structure update gradually converge and the residue sequence-structure consistency is achieved.
>
> Therefore, the two refinement stages are well correlated.
> We will add more discussions of our algorithm in the final version.

---

> ### Comment · Reviewer_rsaq · 2023-08-13
>
> I have read the reply and appreciate the author's reply. My questions are mainly resolved. Nice work.

---

> > ### Author Response · Authors · 2023-08-15
> > **Reply to the reviewer**
> >
> > Thanks for your appreciation and reply! We are glad that our rebuttal resolved your questions.
> >
> > Thank you very much,
> >
> > Authors

---

### Author Rebuttal · Authors · 2023-08-07

**Global response to all reviewers:**

We thank the reviewers for their appreciation and valuable comments! Generally, the reviewers find our paper a novel approach for designing protein pockets that bind to ligand molecules. In the rebuttal, we have done additional experiments and added more discussions and clarifications. With the constructive suggestions from reviewers, we believe our paper will be improved after the rebuttal period!

In the response, we use [1], [2] ... to refer to the reference papers in the original paper and use [r1], [r2] ... to indicate the additional references in the rebuttal.

**Additional response to Reviewer z6bW:**

**Comment 6:** Can you also show Vina docking scores for all the models without re-docking/relaxation?

**Response 6:** Here we show the directly computed Vina scores for all the models without redocking and relaxation (Vina w/o redocking).  For a fair comparison, we exclude the baseline PocketOptimizer here because it uses re-docking and force field relaxation methods in its generation procedure. Generally, the Vina scores w/o redocking and relaxation are higher (lower binding affinity). Our method FAIR can achieve the lowest Vina score on different settings and datasets. It can be attributed to FAIR’s strong capability of pocket sequence-structure co-design. Meanwhile, the coordinates of ligand molecules are updated during the refinement process. Therefore, FAIR relies less on redocking and relaxation.

| Model          | CrossDocked Vina w/o redocking (↓) | CrossDocked Vina(↓) | Binding MOAD Vina w/o redocking(↓) | Binding MOAD Vina(↓) |
|----------------|-------------------|--------------------|---------------------|---------------------|
| DEPACT |-5.820±2.16   | -6.670±2.13|-6.486±2.23 |-7.526±2.05 |
| HSRN |-5.449±2.01    | -6.565±1.95| -5.870±2.04| -7.349±1.93|
| Diffusion |-5.364±1.88     | -6.725±1.83 |-6.358±2.19|-7.724±2.36 |
| MEAN |-5.672±1.79    | -6.891±1.86  | -6.320±1.88|-7.651±1.97|
| FAIR |**-6.365±1.67**| **-7.022±1.75**|**-7.253±1.72** |**-7.978±1.91**|

**Comment 7:**
How do we know the binding pose of the ligand in advance in real-world design scenarios? Do we always need to start with an existing (crystal) structure of the protein-ligand complex?

**Response 7:** FAIR does not require the binding pose of the ligand in advance for pocket design because it is able to refine the ligand coordinates along with the pocket generation process. In practice, we can initialize the ligand poses with chemical tools (e.g., rdkit ) or take existing binding ligand structures as a reference if available [8,45].

---

> ### Comment · Reviewer_z6bW · 2023-08-16
> **Continued response**
>
> >  Our method FAIR can achieve the lowest Vina score on different settings and datasets. It can be attributed to FAIR’s strong capability of pocket sequence-structure co-design. Meanwhile, the coordinates of ligand molecules are updated during the refinement process. Therefore, FAIR relies less on redocking and relaxation.
>
> This is very interesting -- all ML method's Vina score gets worse without re-docking. However, FAIR seems less sensitive to re-docking than other methods. This is good.
>
> However, I am still cautious about the claim that "FAIR’s strong capability of pocket sequence-structure co-design". See my concerns around how FAIR does not improve over the Vina scores for the groundtruth protein-ligand complexes from the dataset.

---

> > ### Author Response · Authors · 2023-08-17
> > **Further Response to Reviewer z6bW**
> >
> > Thanks for the reviewer's valuable reply!
> >
> > **Comment 1:** All ML method's Vina score gets worse without re-docking. However, FAIR seems less sensitive to re-docking than other methods. This is good.
> >
> > **Response 1:** Thanks for the appreciation! It is true that Vina scores get worse without re-docking for all ML methods. In FAIR,
> >  the coordinates of ligand molecules are updated and optimized during the refinement process. Therefore, FAIR is less sensitive to re-docking.
> >
> > **Comment 2:** I am still cautious about the claim that "FAIR’s strong capability of pocket sequence-structure co-design". See my concerns around how FAIR does not improve over the Vina scores for the groundtruth protein-ligand complexes from the dataset.
> >
> > **Response 2:** Due to the diverse distribution of protein-ligand complexes in the datasets and random sampling, it is common that the Vina score distributions have large variance and overlap (see related works such as [r1-r3])
> > Since FAIR is directly trained on the datasets, it is expected that FAIR have similar Vina score distribution of the datasets. For applications in pocket redesign, we can further filter or modify the generated pockets to have a favorable Vina score and other required properties, which is common in real applications [r4-r6].
> >
> > Moreover, we can combine FAIR with optimization methods for better Vina scores. Following [r2], here we leverage the evolutionary algorithm and obtain some favorable results. During the evolutionary algorithm, at the end of every generation the top 10 binding affinity (lowest Vina scores) pockets are used to seed the next population. Every seed pocket is used as the initialization of 20 new candidate pockets with randomly added noise to sequence and structure.
> > After three rounds of iterations, the average Vina scores of the optimized pockets are **-8.632** on CrossDocked and **-9.370** on Binding MOAD, which are obviously better than the test set **-7.016** and **-8.225** respectively. Due to the limited time in the rebuttal period, we will continue to explore applying FAIR for pocket optimization and include the results in our final version.
> >
> > [r1] Peng X, Luo S, Guan J, et al. Pocket2mol: Efficient molecular sampling based on 3d protein pockets, ICML, 2022.
> > [r2] Guan J, Qian W W, Peng X, et al. 3d equivariant diffusion for target-aware molecule generation and affinity prediction. ICLR, 2023.
> > [r3] Schneuing A, Du Y, Harris C, et al. Structure-based drug design with equivariant diffusion models. arXiv preprint arXiv:2210.13695, 2022.
> > [r4] Privett H K, Kiss G, Lee T M, et al. Iterative approach to computational enzyme design. Proceedings of the National Academy of Sciences, 2012, 109(10): 3790-3795.
> > [r5] Xie W J, Asadi M, Warshel A. Enhancing computational enzyme design by a maximum entropy strategy. Proceedings of the National Academy of Sciences, 2022, 119(7): e2122355119.
> > [r6] Polizzi N F, DeGrado W F. A defined structural unit enables de novo design of small-molecule–binding proteins. Science, 2020, 369(6508): 1227-1233.

---

### Decision · Program_Chairs · 2023-09-21

**Decision:**

Accept (spotlight)

**Comment:**

The paper presents an atomic-level sequence and structure design method tailored to protein biding pockets conditioned on the ligand structure.
* The reviewers agreed that the approach is novel, well-written, well-motivated, and well-placed in the context of the literature.
* The proposed method is built from reasonable components found also in other applications of ML to protein design: a coarse-to-fine architecture modeling backbone and residue types first, and side-chains after; as well as the full-shot iterative refinement popular e.g., in antibody design. The work considers also the flexibility of binding ligands, which is a big plus.
* The paper also comes with extensive comparisons and ablations on CrossDocked and Binding MOAD that showcase its benefits over five representative baselines.

I recommend this work for a spotlight presentation as I think that this paper is likely to influence the field of machine learning for protein design.